

# A 30 m annual cropland dataset of China from 1986 to 2021

Ying Tu[1], Shengbiao Wu[2], Bin Chen[2,3,4], Qihao Weng[5], Peng Gong[3,6], Yuqi Bai[1], Jun Yang[1], Le Yu[1], Bing Xu[1,7,*]

[1]Ministry of Education Ecological Field Station for East Asian Migratory Birds, Department of Earth System Science, Institute for Global Change Studies, Tsinghua University, Beijing 100084, China
[2]Future Urbanity & Sustainable Environment (FUSE) Lab, Division of Landscape Architecture, Faculty of Architecture, The University of Hong Kong, Hong Kong SAR, China
[3]Urban Systems Institute, The University of Hong Kong, Hong Kong SAR, China
[4]HKU Musketeers Foundation Institute of Data Science, The University of Hong Kong, Hong Kong SAR, China
[5]Department of Land Surveying and Geo-Informatics, The Hong Kong Polytechnic University, Kowloon, Hong Kong SAR, China
[6]Department of Geography, and Department of Earth Science, The University of Hong Kong, Hong Kong SAR, China
[7]International Research Center of Big Data for Sustainable Development Goals, Beijing, 100094, China

*Correspondence to*: Bing Xu (bingxu@tsinghua.edu.cn)

**Abstract.** Accurate, detailed, and up-to-date information on cropland extent is crucial for provisioning food security and environmental sustainability. However, because of the complexity of agricultural landscapes and lack of sufficient training samples, it remains challenging to monitor cropland dynamics at high spatial and temporal resolutions across large geographical extents, especially for places where agricultural land use is changing dramatically. Here we developed a novel cost-effective annual cropland mapping framework that integrated time-series Landsat imagery, automated training sample generation, and machine learning and change detection techniques. We implemented the proposed scheme to a cloud computing platform of Google Earth Engine and generated China's annual cropland dataset (CACD) at a 30 m spatial resolution for the first time. Results demonstrated that our approach was capable of tracking dynamic cropland changes in different agricultural zones. The pixel-wise F1 scores for annual maps and change maps of CACD were 0.79±0.02 and 0.81, respectively. A further cross-product comparison in terms of accuracy assessment, correlations with statistics, and spatial details indicated the precision and robustness of CACD than other datasets. According to our estimation, from 1986 to 2021, China's total cropland area expanded by 30,300 km$^2$ (1.79%), which underwent an increase before 2000 but a general decline between 2000-2015 and a slight recovery afterward. Cropland expansion was concentrated in the northwest while the eastern coastal region experienced substantial cropland loss. In addition, we observed 419,342 km$^2$ (17.57%) of croplands that were abandoned at least once during the study period. The consistent, high-resolution data of CACD can support progress toward sustainable agricultural use and food production in various research applications. The full archive of CACD is freely available at https://doi.org/10.5281/zenodo.7936885 (Tu et al., 2023a).

**Keywords**. Cropland change detection; Landsat; Training sample; Class probability; LandTrendr; Google Earth Engine; China



## 1 Introduction

The global population has boosted during the past century: from less than 2 billion in 1920 to 7.8 billion in 2020 and is expected to reach 9.7 billion by 2050 (Roberts, 2011). Feeding this growing population, as well as coping with increasing food demand from shifting diets, has become a major challenge for global food security and environmental sustainability (Godfray et al., 2010; Searchinger et al., 2008; Tilman et al., 2011). The United Nations has specified the need for realizing "no poverty and zero hunger" and "balancing increasing agricultural production with the maintenance of ecosystem services" in its sustainable

development goals (SDGs) (United Nations, 2015). Knowledge of where farmlands locate and how they change over time not only facilitates people's understanding of crop area and yield, but also supports early warnings and adaptation initiatives of the agriculture system (See et al., 2015). In this context, detailed, timely, and accurate mapping of cropland extent appears as a critical prerequisite for food market stability, land and water resources management, and the assessment of environmental impacts of agriculture (Waldner et al., 2015b).

As the most populous country on the planet, China has a long history of farming dating back more than 8000 years (Bryan et al., 2018). Today, China feeds about 20% of the world's population with only 7% of the total farmland (Cui and Shoemaker, 2018). This has inevitably placed huge pressure on the country's food security and agricultural sustainability. During the past few decades, extensive croplands were lost in China because of rapid urbanization (Liu et al., 2019; Tu et al., 2021; Tu et al., 2023b). To compensate for the loss, substantial cropland expansion and intensification took place in the meantime (Zuo et al.,

2018), which seriously threatens ecosystem functioning and biodiversity (Tang et al., 2021; Zabel et al., 2019). It is therefore of great importance to track spatial-temporal dynamics of croplands in China to propose more adaptive policies and strategies in the face of dramatically changing agricultural land use.

Earth observations provide a reliable and cost-effective way of long-term, large-scale, and up-to-date cropland information gathering. The Moderate Resolution Imaging Spectroradiometer (MODIS) imagery, for example, provides a unique capability

to map cropland extent at 250-500 m spatial resolutions (Gumma et al., 2014; Wardlow and Egbert, 2008; Xiong et al., 2017; Zhang et al., 2022; Zhang et al., 2015). However, such kinds of data may fail to detect small field patches (<2.56 ha) (Fritz et al., 2015), especially for heterogeneous agricultural landscapes. In recent years, higher-resolution data like Landsat and Sentinel-2 have been well used in global cropland mapping initiatives at tens-of-meter levels, including Finer Resolution Observation and Monitoring-Global Cropland (FROM-GC) (Yu et al., 2013), Global Food Security-Support Analysis Data

(GFSAD30) (Thenkabail et al., 2021), and Global Land Analysis & Discovery Cropland Data (GLAD) (Potapov et al., 2022). These are often one-phase or multi-year cropland data products. In some developed countries and regions, more detailed cropland maps, such as the United States National Agricultural Statistics Service Cropland Data Layer (NASS-CDL) (Boryan et al., 2011) and the European Union 10 m crop type map (D'andrimont et al., 2021), have been produced. To date, no fine-resolution annual cropland dataset of China exists yet.

Monitoring cropland dynamics is demanding given the high inter- and intra-annual spectral variabilities of cropland (Prishchepov et al., 2012; Yin et al., 2014). From the perspective of mapping schemes, current approaches undergo a shift





from multi-year cropland classification with traditional machine learning to annual cropland mapping that combines change detection techniques. In the former strategy, cropland is usually considered as a general land use category, of which classifications are performed individually for each year using in situ data and supervised classifiers such as decision tree

(Boryan et al., 2011; Pittman et al., 2010; Potapov et al., 2022), random forest (Li et al., 2022; Yang and Huang, 2021; Yin et al., 2020), and support vector machine (Lambert et al., 2016; Shao and Lunetta, 2012; Traoré et al., 2019). However, the accuracy of classification maps is highly dependent on data quality, and errors present in each of the maps are compounded in the final change map, which may impair its ability in capturing real cropland use transitions (Zhu, 2017). To overcome this issue, post-classification processing methods, such as consistency checking (Li et al., 2015) or spatial-temporal filtering (Liu

and Cai, 2012), have been employed to increase mapping accuracy. Instead, the latter strategy leverages all time-series information of remote sensing imagery into a change detection model and has become increasingly popular. Normally, temporal features of spectral bands or indices are used to fit temporal trajectories to identify both abrupt changes and continuous trends (Zhu, 2017). Representative change detection algorithms include Landsat-based detection of Trends in Disturbance and Recovery (LandTrendr) (Kennedy et al., 2010), Breaks for Additive Season and Trend (BFAST) (Verbesselt

et al., 2010), and Continuous Change Detection and Classification (CCDC) (Zhu and Woodcock, 2014), of which trials have been given to characterize cropland patterns (Schneibel et al., 2017; Zhu et al., 2019). Nevertheless, since most of these algorithms are not originally developed for cropland change detection, the result can be highly variable and is often mixed with other vegetation types such as forest and grassland (Pasquarella et al., 2022).

One possible solution to deal with the shortcomings mentioned above is to take the advantages of both strategies, that is, to

first perform supervised classifications to generate multi-epoch land cover maps or cropland probabilities and then apply this intermediate information as inputs for change detections. A few efforts have been made in this manner so far, but challenges remain, including (1) limited exploration of the time-series sequence. Change detection analyses are simply used as a final step for updating the year of change (Dara et al., 2018; Xu et al., 2020; Xu et al., 2018), while the rich information stored in fitted trajectories of temporal segmentation is not well-incorporated in cropland distinguishing. (2) Unreasonable settings of land

use conversion rules. Xu et al. (2020), for instance, assumed there was a low probability of frequent land use changes and therefore only detected one single change during each 5-year interval. However, this is often not the case with croplands. (3) Most studies are conducted for specific type of cropland use change (e.g., cropland abandonment) and tested in small areas (Dara et al., 2018; Yin et al., 2018; Xu et al., 2021). It remains unclear whether the mapping scheme can be applied to monitor large-scale cropland dynamics.

Training samples are another essential issue in effective cropland classification. Existing studies rely extensively on in situ data or human interpretation of spectral signatures, making the classification process resource-intensive, time-consuming, and difficult to repeat over space and time (Zhong et al., 2014). To cope with the constrain, recent initiatives have been made to extract labeling or training data from existing land cover information (so called "baseline maps") (Huang et al., 2020; Radoux et al., 2014; Waldner et al., 2015a; Xie et al., 2019; Zhang and Roy, 2017). Zhang and Roy (2017) derived training samples of

classification from previous 500 m MODIS land cover products. Huang et al. (2020) proposed an automatic training sample



migration method by measuring the spectral similarity and spectral distance between the reference spectra and image spectra. These achievements pave a new way for mapping annual croplands cost-effectively.

The aim of this study is to propose a novel paradigm for large-scale fine-resolution cropland dynamics monitoring. We first generated training samples automatically based on prerequisite knowledge, existing land cover baseline maps, and a time-105 weighted dynamic time warping method. We then input multi-year phenological features derived from time-series Landsat imagery into a random forest classifier to obtain per-pixel annual cropland probabilities. By further applying the LandTrendr change detection algorithm, we classified the cropland probability time series into several segments, in which breakpoints and corresponding years of changes were recorded. Here we inspected cropland changes as a continuous dynamic process that consisted of several trajectories including no change, cropland gain, or cropland loss. We established a set of rules to 110 discriminate annual cropland types from the LandTrendr segmentation results. Based on the proposed framework, we produced the first annual cropland dataset of China (CACD) for 1986-2021 at a 30 m spatial resolution. We examined the accuracy of CACD using independent validation sample sets and compared the results with existing cropland maps. Finally, we assessed spatial and temporal changes of croplands in China since 1980s with CACD.

## 2 Methodology

As the framework delivered in Fig.1, we have proposed a trajectory-based approach that combines machine learning and change detection techniques for mapping annual cropland dynamics. Annual cropland in this study is defined as a piece of land of 0.09 ha in minimum (minimum width of 30 m) that is sowed/planted and harvestable at least once within the 12 months after the sowing or planting date. This definition is consistent with the Joint Experiment of Crop Assessment and Monitoring (JECAM) network (Defourny et al., 2014), which adopts a shared scope of cropland that meets FAO's Land Cover Meta 120 Language (Di Gregorio, 2005). Three exceptions are excluded in the annual cropland definition: (1) sugarcane plantation and cassava crop, as they have a longer vegetation cycle and are not yearly planted; (2) small plots such as legumes that do not meet the minimum size criteria of cropland; and (3) greenhouse crops, as their signals in remote sensing images are very different from others.

China, a traditional agricultural nation in east Asia, is chosen as the study area for testing and verifying the proposed approach 125 under large scales and dynamic climates. The country shares a total area of 9,600,000 km$^2$, including mainland China, Hong Kong, Macau, and Taiwan. According to the national comprehensive agricultural zoning promulgated by the Chinese government, our study area consists of nine agricultural zones in general (Fig. S1). Their geographical and climatic conditions, growing seasons, cropping patterns, and major crops are provided in Table S1. Considering local differences and computation capacity, we further divide the study area into around 1700 subregions with a size of 0.8°×0.8° (Fig. S1). Within each 130 subregion, we perform annual cropland classifications individually on Google Earth Engine (GEE, https://earthengine.google.com/) based on the proposed methods in Fig. 1, of which details are described in the following Sections 2.1-2.7.



## 2.1 Training data generation

Training samples emerged as a key component in supervised classification. Here we developed an effective approach for automated training samples generation using existing land cover maps and the time-weighted dynamic time warping (TWDTW) method. First, we built a Cropland Inspector Tool on GEE that depicted time-series NDVI profiles and corresponding Landsat imagery for a given point of interest. An illustration of the interpretation process of annual cropland types was given in the supplementary materials Fig. S2. Based on the tool, we manually interpreted 100 stable cropland points and 100 non-cropland (including forests, water ponds, and impervious areas) points within each agricultural zone as the referencing set. Second, we overlapped two annual land cover datasets in China, CLCD (Yang and Huang, 2021) and CLUD (Xu et al., 2020), to extract cropland and non-cropland masks for each year respectively. For each subregion, we produced thousands of potential cropland and non-cropland points randomly and compared their monthly NDVI sequences with those of the referencing set using TWDTW, an algorithm that had been developed for distinguishing between different types of land use and land cover (Maus et al., 2016). TWDTW works by comparing the similarity between two temporal sequences and finds their optimal alignment, resulting in a dissimilarity measure (Belgiu and Csillik, 2018). Based on the discrimination results, we retained 20% invariant samples that had the lowest dissimilarity value compared to the referencing set. The threshold value was set following recommendations by Ghorbanian et al. (2020). To determine how many training samples were least required for a robust classification, we conducted multiple experiments with different training sizes. As shown in Fig. S3, the F1 score of cropland classification increased as the number of training samples grew in the initial stage but became rather stable after reaching 800. Consequently, we generated ~800 training samples within each subregion.

## 2.2 Feature space construction

We used all available Landsat TM/ETM+/OLI Tier 1 surface reflectance imagery from 1986 to 2021 covering our study area as of March 2022. The data had been atmospherically corrected by USGS using the LEDAPS (Masek et al., 2006) and LaSRC (Vermote et al., 2016) algorithms. We further masked out all the clouds, cloud shadows, and snow/ice contained in the Landsat imagery using the FMASK (Zhu and Woodcock, 2012). Considering the differences between sensors, we applied the coefficients suggested by Roy et al. (2016) to normalize OLI reflectance to that of TM and ETM+. This enabled multi-temporal images to be consistent and comparable across years. We then calculated spectral indices of NDVI, NDSI, NDBI, NBR, and EVI based on the formulas provided in Table S2. To improve classification performance, we finally carried out the tasseled cap transformation to include Brightness, Greenness, and Wetness indices using the coefficients provided by Crist (1985).

Multi-temporal metrics have been widely used for capturing the spatial-temporal features of different land covers, such as forests, croplands, and urban areas (Potapov et al., 2012; Yin et al., 2020). Due to the nature of crop growth, spectral signals of the croplands exhibit phenological patterns periodically. An example is shown in Fig. S4, where NDVI is high during the growing season and low during sowing. Accordingly, based on the processed Landsat data, we extracted a series of multi-temporal metrics each year as input features for annual cropland classification. These included the $10^{th}$, $25^{th}$, $50^{th}$, $75^{th}$, and $90^{th}$



percent quantiles of both growing and non-growing seasons for the spectral bands and indices listed in Table S2. In addition, we augmented elevation information into feature space using the Shuttle Radar Topography Mission (SRTM) digital elevation dataset (Jarvis et al., 2008). This resulted in a total of 141 input features for each year.

### 2.3 Cropland probability estimation

We estimated pixel-wise cropland probabilities for each year between 1986-2021 using the random forest algorithm. Random
forest is a bagging ensemble learning approach that constructs a multitude of decision trees at the training time (Ho, 1995). The final class (classification) or prediction (regression) is assigned based on the majority "vote" of all trees (Breiman, 2001). The algorithm has been widely used in land cover and land use classifications, given its accuracy and efficiency in dealing with high-dimensional data (Gong et al., 2019; Tu et al., 2020). In this study, the parameter for the number of trees (*numberOfTrees*) was set to 500, the number of variables per split (*variablesPerSplit*) was set to the square root of
*numberOfTrees*, and other parameters were set as default as those on GEE.

### 2.4 Temporal segmentation

We employed the trajectory-based spectral-temporal segmentation algorithm, LandTrendr, to decompose the pixel-wise annual cropland probability time series into different cropland use segments. The overarching goal of LandTrendr is to characterize a temporal trajectory of data values using a sequence of connected linear segments bounded by breakpoints known as "vertex"
(Kennedy et al., 2018). By fitting a series of linear segments, LandTrendr reduces inter-annual signal noise while capturing both long-term gradual and short-term drastic changes (Kennedy et al., 2010). The fitting result of LandTrendr is a series of segments composed of breakpoints that separate periods of durable change or stability in spectral trajectory (Fig. 2). In this way, the magnitude of changes as well as years when changes occur can be recorded. LandTrendr was originally developed for detecting forest disturbances, but recent applications also demonstrated its potential in monitoring cropland dynamics (Dara
et al., 2018; Yin et al., 2018).

The implementation of LandTrendr requires a series of control parameters and filtering steps designed to reduce overfitting while still capturing the desired features of trajectories. Nevertheless, when leveraging LandTrendr to monitor cropland dynamics, we found few experiences with parameter settings in the literature. In addition, differences in climatic, geographic, and cropping conditions across regions would also affect algorithm performance. Accounting for these issues, we defined one
test region for each agricultural zone with a size of 100 km×100 km (Fig. S1). Within each test region, we interpreted 100 cropland and non-cropland samples likewise, and compared classification results under different LandTrendr arguments using the developed approach (Table S3). Practically, we assigned LandTrendr parameter settings with the highest classification accuracy of F1 score as the input for annual cropland mapping in the corresponding agricultural zone (Table S4). Details are given in the supplementary materials.



## 2.5 Annual cropland mapping

We have established a set of rules that automatically discriminate cropland transition types based on the LandTrendr segmentation results of annual cropland probability time series. Suppose each segment represents a specific cropland use status or change, the key is to determine its transition type characterized by the fitting breakpoints in the beginning and end, and then to reverse cropland types for each year within the segment. Take the pixel in Fig. 2 as an example, LandTrendr has divided its time series of cropland probabilities into three end-to-end segments ($s$) and four breakpoints ($p$), where $t$ and $v$ represent the year in which cropland use changes occur and the fitted cropland probability. First, land use conditions for all breakpoints $p_{(t,v)}$ are identified based on the fitted value: cropland if $v > 0.5$ and non-cropland if vice versa. Since each segment $s_i$ is delineated by a starting breakpoint $p_{is}$ and an ending breakpoint $p_{ie}$, this results in four scenarios for the transition type of $s_i$ (Table 1). If both $p_{is}$ and $p_{ie}$ are cropland (non-cropland), land use types of the pixel during $t_{is}$-$t_{ie}$ remain unchanged and should always be cropland (non-cropland). If $v_{is}>0.5$ and $v_{ie}\leq0.5$, it means cropland loss takes place during the period. In this case, the pixel will be recognized as cropland in $t_{is}$ and non-cropland from $t_{is+1}$ to $t_{ie}$. Conversely, under the scenario where $p_{is}$ is non-cropland and $p_{ie}$ is cropland, land use types of the pixel are assigned as cropland in all years of $s_i$ except for $t_{is}$ (which is non-cropland).

## 2.6 Post-classification processing

In post-classification processing, we first applied a 2-D Gaussian low pass filter to the classified cropland maps. The Gaussian filter is one of the most widely used and effective window-based filtering methods for reducing noise and blurring regions of an image (Canny, 1986). Weights for each pixel in the filtering window are determined by a Gaussian function:

$$G_{x,y} = \frac{1}{2\pi\sigma^2} e^{-(x^2+y^2)/2\sigma^2} \ , \tag{1}$$

where $x$ and $y$ are the coordinates of the pixels in the filtering window. The coordinate of the center pixel of the window is (0,0). $\sigma$ is the standard deviation of the Gaussian filter that controls the degree of blurring spectral information. The filtering window is also controlled by a kernel, which determines the neighborhood extent that can be computed on the central pixel (usually odd numbers). In this study, a 3×3 Gaussian filter was performed using the OpenCV package in Python.

After filtering, a spatial-temporal consistency check approach proposed by Li et al. (2015) was applied to refine the annual cropland maps. Furthermore, we used the JRC global surface water (Pekel et al., 2016) and global artificial impervious area (GAIA) (Gong et al., 2020) datasets to mask out permanent water and impervious areas in each corresponding year respectively.

## 2.7 Accuracy assessment and inter-comparison

Map confidence is a key reflection of classification performance and data quality. In this study, three independent sample sets were used to evaluate the accuracy of CACD comprehensively. The first one was the annual validation sample set generated by the research team. Specifically, we divided the study area into nearly one thousand hexagons and randomly created five points within each hexagon, following the work by Gong et al. (2013) on sample design. For each random point, land cover



status and changes were interpreted manually. This was realized based on the Cropland Inspector Tool and historical high-resolution images from Google Earth. Six well-trained researchers were involved in the interpretation task. Each validation point would be identified by at least two researchers, of which yearly land cover types between 1986-2021 (cropland or non-cropland) as well as confidence level (low or high) were filled in the result sheet. If there was a disagreement between the interpretation results, a final judgment would be made through discussions among the research team. We discarded points with low confidence and retrieved a total of 4972 validation samples by the end (Fig. 3).

Moreover, we leveraged two third-party sample sets (namely GLCVSS and GeoWiki) for the accuracy assessment of CACD. The global land cover validation sample set (GLCVSS) was the first all-season validation sample set generated for global land cover mapping (Li et al., 2017). It contained ~35000 validation samples interpreted on Landsat 8 with records of the date of image acquisition (interpreted from 2013 to 2015), spectral reflectance for each season, and level of interpretation uncertainty and sample sizes (3×3, 9×9, 17×17, 33×33, 1 unit =30 m). The classification system of GLCVSS was the same as FROM-GLC. We selected all GLCVSS points in China with high confidence and a sample size of 30 m×30 m (Fig. S5). The other sample set Geo-Wiki was a global cropland reference dataset collected through a crowdsourcing campaign, which enveloped 80 participants from around the world to review almost 36,000 sample units during September 2016 (Laso Bayas et al., 2017). Each record in the Geo-Wiki dataset represents a cropland site identified by different participants. Likewise, we extracted all the cropland points located in China from Geo-Wiki (Fig. S6). In summary, there were 2018 GLCVSS and 1674 Geo-Wiki samples included in the research.

A group of accuracy metrics was calculated based on the confusion matrix, including F1 score, overall accuracy (OA), producer's accuracy (PA), user's accuracy (UA), and Kappa coefficient. The F1 score was used as a major indicator of accuracy as it conveyed the balance between UA and PA, which reached its best value at 1 and worst value at 0 (Chen et al., 2021). Accuracy metrics of both yearly maps and change maps of CACD were calculated based on the annual validation sample set while GLCVSS was used for assessing the accuracy of CACD in 2015. In terms of Geo-Wiki, because it only contained cropland samples, we calculated the agreement level between Geo-Wiki and CACD in 2016, that is, how many points of GeoWiki were classified as cropland in CACD.

In addition, we compared CACD with other publicly available land cover or cropland datasets (i.e., CLCD, CLUD, GLAD, and GFSAD). Both CLCD (Yang and Huang, 2021) and CLUD (Xu et al., 2020) were 30 m annual land cover datasets in China with a long time span. For each year in CLCD and CLUD, we extracted cropland extent (1) and grouped other land cover types as 'non-cropland' (0). GLAD represented a globally consistent cropland extent time series at 30 m spatial resolution and was mapped in four-year intervals (2000-2003, 2004-2007, 2008-2011, 2012-2015, and 2016-2019) (Potapov et al., 2022). GFSAD was a 30 m global cropland dataset for 2015 (Thenkabail et al., 2021). We calculated accuracies of CLCD, CLUD, GLAD, and GFSAD using the three validation sample sets. Besides, we estimated the total area of croplands at the provincial scale and compared it with statistics.



## 3 Results

### 3.1 Accuracy assessment of CACD

We performed pixel-wise accuracy assessment of CACD based on the annual validation sample set. The F1 score, OA, UA, PA, and Kappa coefficient of annual maps on average were 0.79±0.02, 0.93±0.01, 0.79±0.01, 0.79±0.02, and 0.75±0.02, respectively. Temporally, mapping accuracy after 2000 was generally higher than that before (Table S5), probably due to the improvement of data quantity and quality of Landsat images in the later period. Besides, we estimated mapping accuracy of CACD at hexagon and province levels using the annual validation samples set. Spatially, except for parts of the western and

southeastern coastal areas, the mean F1 score for most of the study area was above 0.8, with omission and commission errors less than 0.25 (Fig. 4a-c). When it comes to provincial performance, the mean F1 score for Northeast Plain, North China Plain, Jiangsu, Zhejiang, Jiangxi, and Yunan reached up to 0.8, whereas a lower value was found in inland provinces such as Inner Mongolia and Xinjiang (Fig 4d). The relatively high omission error in Qinghai and Fujian (Fig. 4e) demonstrated an underestimation of cropland in these regions. In contrast, Guangdong had the highest commission error on average (Fig. 4f),

indicating that croplands were somewhat overestimated in the province.

In terms of change accuracy, we reclassified the annual validation sample set as changed if any cropland use conversion was identified between 1986-2021 and vice versa. Since the number of non-changed samples (98%) was substantially greater than that of changed samples (2%), we randomly subset 100 non-changed samples and combined them with changed samples for further evaluation. Results demonstrated that the F1 score, OA, and Kappa coefficient of CACD change maps could be as high

as 0.81, 0.84, and 0.68 (Table 2). Accuracy for the year of change ranged between 0.76-0.87 as the tolerance years increased from 0 to ±5 years (Table S6).

Furthermore, using two third-part sample sets, the F1 score of CACD in 2015 was verified as 0.82 based on the GLCVSS sample set (Table 3), while 86% of the sample points in Geo-Wiki were correctly classified as cropland in CACD (Table 4). Evaluation results of third-party sample sets complemented the accuracy of CACD.

### 3.2 Comparison with other products


We compared CACD with four existing land cover (cropland) datasets (CLCD, CLUD, GLAD, and GFSAD) at 30 spatial resolutions from multiple aspects. CACD was superior to other products in all comparison years in terms of mapping accuracy (Fig. 5). In 2015, for example, the F1 score of CACD was 0.85, followed by GFSAD (0.77), GLAD (0.76), CLCD (0.76), and CLUD (0.69). Similar results were observed in the accuracy assessment results of GLCVSS and Geo-Wiki sample sets (Tables

3-4). Spatially, the five products were consistent in delineating cropland distributions in major production areas including the Northeast Plain, North China Plain, Sichuan Basin, and Xinjiang (Fig. 6). However, differences occurred in Loess Plateau, Yunan-Guinan Plateau, and the vast hilly regions along the southeast coast. Overall, according to the frequency that how often pixels were identified as cropland, 65.62% of the study area were labeled as croplands at least three times out of the five products (Fig. 6, matched level >= 3).





Fig. 7 compares spatial details of cropland distribution of the five products in six selected regions. In the Northeast Plain (region A), CACD, GLAD, and GFSAD depicted the extent of cropland well, while CLCD slightly overestimated and CLUD severely underestimated croplands. In Xinjiang and North China Plain (region B and C), high spatial consistency was observed between the five products, with CACD and GLAD displaying more details (like the road network). Classification differences mainly appeared in the southern hilly areas (region D, E, and F). In general, CACD, CLCD, and CLUD were more consistent

and closer to the actual cropland distribution. For instance, region D was a terracing land in Sichuan Province, southwestern China. The region was mainly used for rice cultivation, yet those croplands were massively underestimated in GLAD and GFSAD.

Using the five products, we counted cropland area at the provincial scale separately and compared it with statistical data in 2015. As shown in Fig. 8, CACD showed the best agreement with statistics ($R^2$=0.93), followed by CLUD ($R^2$=0.91), CLCD

($R^2$=0.89), GFSAD ($R^2$=0.87), and GLAD ($R^2$=0.74). The scattered points between CLCD and statistics were basically above the 1:1 diagonal, indicating an overestimation of cropland area. On the contrary, GLAD had an underestimated cropland area as more scattered points were located below the 1:1 diagonal. This was mainly caused by GLAD's misclassification in southern China. Taking together, the comparison results between different products demonstrated the accuracy of our annual cropland datasets.

**4 Discussion**

**4.1 Cropland dynamics in China**

Based on the produced CACD, we calculated provincial net changes in cropland area between 1986-2021 and plotted annual cropland area dynamics at the national scale. We also employed the annual validation samples to an error-adjusted stratified estimation approach following Olofsson et al. (2014) to estimate sample-based cropland area and its 95% confidence interval.

As shown in Fig. 9a, cropland gain mainly occurred in northern China, including Xinjiang, Inner Mongolia, and Heilongjiang. In Xinjiang, for example, 44635 km$^2$ of the land area (~3%) was developed as cropland during the past 36 years, a figure that was equivalent to 1.3 times the area of Hainan Province. On the contrary, most provinces in eastern, central, and southern China experienced massive cropland loss. The top five provinces with the largest area of cropland loss were Sichuan (12739 km$^2$), Jiangsu (10865 km$^2$), Shaanxi (10700 km$^2$), Hebei (8566 km$^2$), and Shandong (8407 km$^2$). In the capital city of Beijing,

more than 2000 km$^2$ croplands were occupied during the study period, which accounted for 12% of the city's total area. Additionally, cropland areas in some inland provinces (such as Guizhou) remained rather stable. Overall, according to sample-based estimation, China's cropland area increased from 1,694,900 ± 193,700 km$^2$ in 1986 to 1,725,200 ± 212,400 km$^2$ in 2021, a net increase of 30,300 km$^2$ (1.79%). Temporally, the country's cropland area increased before 2000 but gradually decreased until 2015, after which a recovery phase began (Fig. 9b). These findings are generally consistent with those reported in previous

empirical studies (Liu et al., 2014a; Gao et al., 2019).



Rapid urbanization process in China since the 1980s has caused significant losses of cropland, particularly in large cities and the eastern coastal region (Qiu et al., 2020; Tu et al., 2021; Tu et al., 2023b). In response to the increasing demand for food, the Chinese government has implemented a series of farmland protection policies, including the Requisition–Compensation Balance of Farmland (RCBF) policy, which mandates that any lost cropland must be offset by the expansion of cropland elsewhere (Liang et al., 2015; Liu et al., 2014b). Despite these efforts, croplands in China have undergone dramatic and unbalanced changes over the past few decades. While the net change in cropland area has been relatively small, the spatial-temporal variance is noteworthy (Gao et al., 2019; Liu et al., 2014a; Yu et al., 2018). Fig. 10 showcases four typical types of cropland dynamic change in China with CACD. In the Ar Horqin Banner of Chifeng city, Inner Mongolia, large-scale croplands were developed for pasture reclamation and cultivation during the past decades (Fig. 10a). Similarly, vast agricultural land parcels sprang up in Aksu, Xinjiang for cotton cultivation (Fig. 10b). In Shanghai, China's most significant metropolitan, remarkable socioeconomic development since the reform and opening up in the 1980s had led to extensive urban sprawl, encroaching substantial areas of pre-existing croplands (Fig. 10c). In the mountainous area of Tongren, Guizhou, some croplands were abandoned in recent years as a result of urban-rural migration and declining marginal benefits of arable land (Fig. 10d). In summary, these specific examples demonstrated that CACD was capable of capturing fine-resolution cropland dynamics across different landscapes.

Cropland abandonment is a widespread and rapidly increasing phenomenon worldwide (Queiroz et al., 2014), as a result of land marginalization caused by unsatisfactory land suitability and reduced economic viability (Macdonald et al., 2000). It has wide-ranging effects on the environment and social economy, such as biodiversity conservation (Isbell et al., 2019), carbon storage (Vuichard et al., 2008), and land economics (Ito et al., 2016). However, knowledge of the extent and timing of cropland abandonment in China is lacking. According to FAO, cropland abandonment refers to land that has not been cultivated for at least five consecutive years (Fao, 2016). Following this definition, we identified a pixel of CACD as abandoned if it was classified as cropland in the initial year but changed to non-cropland for the following five years. Also, we were conservative and included only those croplands as abandoned that converted to natural cover, not those to built-up lands. To do so, we masked out cropland conversions to impervious areas based on the GAIA data. When abandonment occurred, we labeled its timing as the first year in which cropland was no longer active (starting from 1990 to 2015).

Our results revealed that 419,342 km$^2$ (17.57%) of the croplands in China were abandoned at least once during the past decades. Most abandoned croplands were distributed in the central and the west (from 100° to 125°E and from 20° to 50°N, Fig. 11a), particularly in Inner Mongolia, Gansu, Shaanxi, and Yunan provinces (Fig. S7). This spatial pattern was consistent with Li et al. (2018), who estimated the extent of cropland abandonment in China's mountainous regions using multiple regression models. The average altitude of abandoned cropland was 860 m, 81% of which were in regions below 1500 m (Fig. 11b). Temporally, the annual cropland abandonment area in China showed an upward trend from 1990 to 2015, which increased from 7516 km$^2$ to 14823 km$^2$ (Fig. 11c). Rural-urban migration, reductions in agricultural labor forces, and rural population ageing have been invoked as key drivers of cropland abandonment in China (Zhang et al., 2014; Yan et al., 2016; Ren et al., 2023).



## 4.2 Strengths and potential implications

This research introduces a new framework for monitoring annual cropland dynamics at the 30 m spatial resolution. Our methods offer several contributions to the existing literature. First, we leverage baseline land cover maps and the TWDTW discrimination algorithm to realize automated training sample generation. This eliminates the time-consuming and labor-costive process of traditional training sample collection, enabling cost-effective cropland mapping at large scales. Second, we adopt the random forest classifier for annual cropland probabilities estimation and then integrate time series of these probabilities as spectral metrics into LandTrendr for cropland trajectory modeling. The incorporation of machine learning and change detection techniques not only increases accuracy but also improves the spatial-temporal consistency of classification results. Third, we establish a set of transition rules to convert the LandTrendr fitting results to pixel-wise annual cropland types, which can capture various cropland use changes such as abandonment or fallow. This novel strategy is distinct from existing initiatives (Potapov et al., 2022; Dara et al., 2018; Xu et al., 2018; Yin et al., 2020). Our results highlight the potential of change detection algorithms like LandTrendr to complement traditional classification processes in identifying dynamic land cover changes effectively.

In practice, we partitioned the study area into nine agricultural zones and performed localized annual cropland mapping within each 0.8°×0.8° subregion. We also evaluated the impacts of training sample size and LandTrendr parameter settings on classification accuracy. Localized classifications may greatly improve the accuracy in heterogeneous regions such as southern China. Our experiments provide valuable insights for future land cover and land use mapping endeavors. Moreover, we took advantage of the powerful data storage, computing, and analysis capabilities of GEE to build an end-to-end framework, which enables fast annual cropland mapping in any given area of interest worldwide. Theoretically, the proposed framework is highly adaptable and can be extended to map other land use types.

Based on the aforementioned mapping scheme, we produced the first 30-m annual cropland maps of China for the period 1986-2021. The accuracy of the dataset was validated through three independent validation sample sets and multi-perspective comparisons with other products. The CACD dataset is a valuable contribution to the field of agriculture and land use management in China. Its accuracy and high-resolution information provide insights into the changing dynamics of cropland use over time, helping inform policymakers and stakeholders in developing sustainable land use practices. Our subsequent analyses with CACD revealed a dramatic but heterogeneous change in cropland dynamics in China, with over a third of croplands experiencing at least one change of land use during the study period. Furthermore, we identified a total of 419,342 km² (17.57%) of abandoned croplands from 1990 to 2015, mostly located in the central and western mountainous areas. With the ability to detect and analyze cropland abandonment patterns, targeted interventions can be implemented to bring abandoned land back into use, promoting food security and protecting arable land resources. Additionally, CACD can be utilized to conduct agriculture-related studies, such as analyzing crop yield and productivity, assessing the impact of climate change on crop growth, and monitoring land use changes over time. Overall, CACD represents a powerful tool for promoting sustainable agriculture practices and ensuring the long-term availability of arable land resources in China.





## 4.3 Limitations and prospects

A few limitations regarding methodology and data are included in the study. First, we monitor cropland dynamics at an annual
time span but do not consider intra-annual variations of crops (e.g., different crop calendars and cropping intensity).
Consequently, our results may not be effective in identifying perennial crops. Second, we depict the general extent of croplands
but do not differentiate specific types. This becomes essential when evaluating yields or climate responses of different crops.
Future work needs to explore advanced methods for fine-resolution crop types mapping. Third, despite the high accuracy,
CACD is subject to several levels of uncertainty. Temporally, CACD has a relatively low accuracy before 2000, given the
uneven coverage of Landsat 5 data. Spatially, mapping accuracy varied across different regions, where the western and
southeastern coastal areas have a comparatively high error compared to others. This may be caused by the complex composition
of local landscapes and the low quality of Landsat data (e.g., unmasked clouds). Besides, the segmentation results of
LandTrendr can be influenced by small changes in spectral inputs and key parameters (Table S4). We recommend users be
careful when applying CACD in these regions.

## 5 Data availability

The atlas of CACD generated in this study can be accessed at https://doi.org/10.5281/zenodo.7936885 (Tu et al., 2023a). All
maps are at the 30 m spatial resolution under the EPSG:4326 (WGS84) spatial reference system.

## 6 Conclusions

Timely and dependable cropland distribution maps appear as key variables for crop yield estimation, agricultural sustainability
assessment, and climate modeling. This research put forward a cost-effective fine-resolution cropland dynamics monitoring
scheme by synthesizing automated training sample generation, random forest supervised classification, and the LandTrendr
temporal segmentation algorithm. We showed how labor on training sample collection can be substantially reduced by making
full use of existing land cover datasets. We also demonstrated how state-of-art machine learning and change detection
approaches can be synthesized to characterize cropland dynamics. Leveraging the full archive of Landsat imagery and the GEE
cloud computing platform, we mapped the first annual cropland distributions in China at a 30 m resolution from 1986 to 2021.
The generated dataset reached a promising accuracy of F1 score of 0.79±0.02, which was superior to other products including
CLCD, CLUD, GLAD, and GFSAD. Furthermore, validations of third-party sample sets, regressions with provincial statistics,
as well as comparisons of spatial details between multiple products, indicated that CACD was reasonable in delineating spatial
distributions and temporal trends of cropland dynamics. The total cropland area of China in 2021 was 1,725,200 ± 212,400
km$^2$, which increased by 30,300 km$^2$ (1.79%) compared to that in 1986. Cropland expansion mainly took place in northwestern
China while the eastern coastal region withstood substantial cropland loss due to rapid urbanization. The annual cropland
abandonment area was 16,128 km$^2$ on average and showed an increasing trend from 1990 to 2015. The fine-resolution annual

cropland data of this study is expected to serve as the basis for a wide spectrum of applications and decision makings in the future, for example, facilitating crop condition monitoring and early warning, promoting progress towards sustainable food
production, and assessing the environmental impacts of agricultural expansion and intensification.

**Author contributions**

Ying Tu and Bing Xu conceived the research idea. Ying Tu designed the study, created the dataset, carried out the analysis, and wrote the first draft of the manuscript, under the supervision of other authors. All authors participated in the review and editing of the manuscript.

**Competing interests**

The authors declare that they have no conflict of interest.

**Acknowledgements**

The authors would like to express their great gratitude to the following organizations for their contributions to this research: USGS for providing free access to Landsat imagery and GFSAD data, Wuhan University for providing CLCD data, Tsinghua
University for providing CLUD and GLCVSS data, the University of Maryland for providing GLAD data, and Geo-Wiki for sharing the third-party validation samples. We thank Prof. Jian Xu and students from Jiangxi Normal University for their efforts in sample collection. We also thank Prof. Jinwei Dong and Prof. Li Wang from the Chinese Academy of Sciences for their valuable comments on the manuscript.

**Financial support**

This study was supported by the Open Research Program of the International Research Center of Big Data for Sustainable Development Goals (CBAS2022ORP02), National Key Research and Development Program of China (2022YFB3903703, 2022YFE0209300), Science and Technology Commission of Shanghai Municipality (22dz1209602)

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



**Figures**

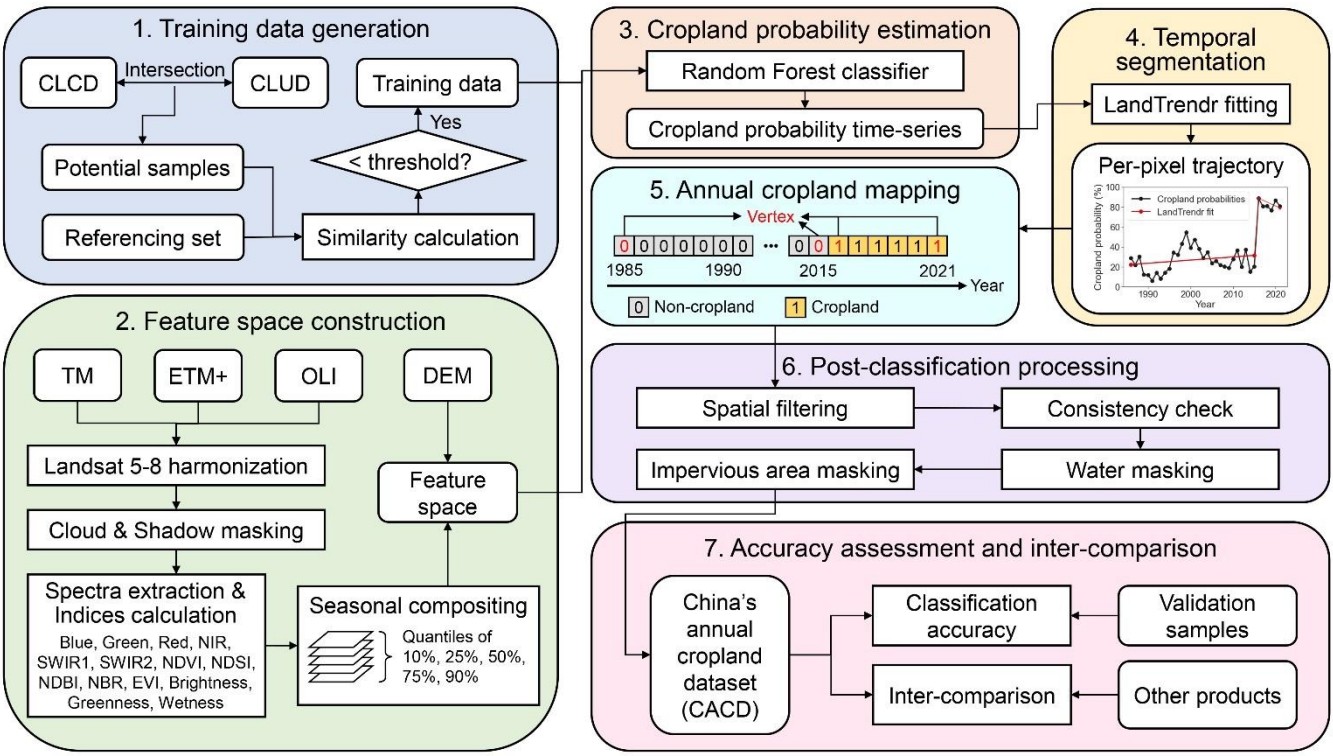

700             **Figure 1: Flowchart of this study for mapping annual cropland dynamics.**

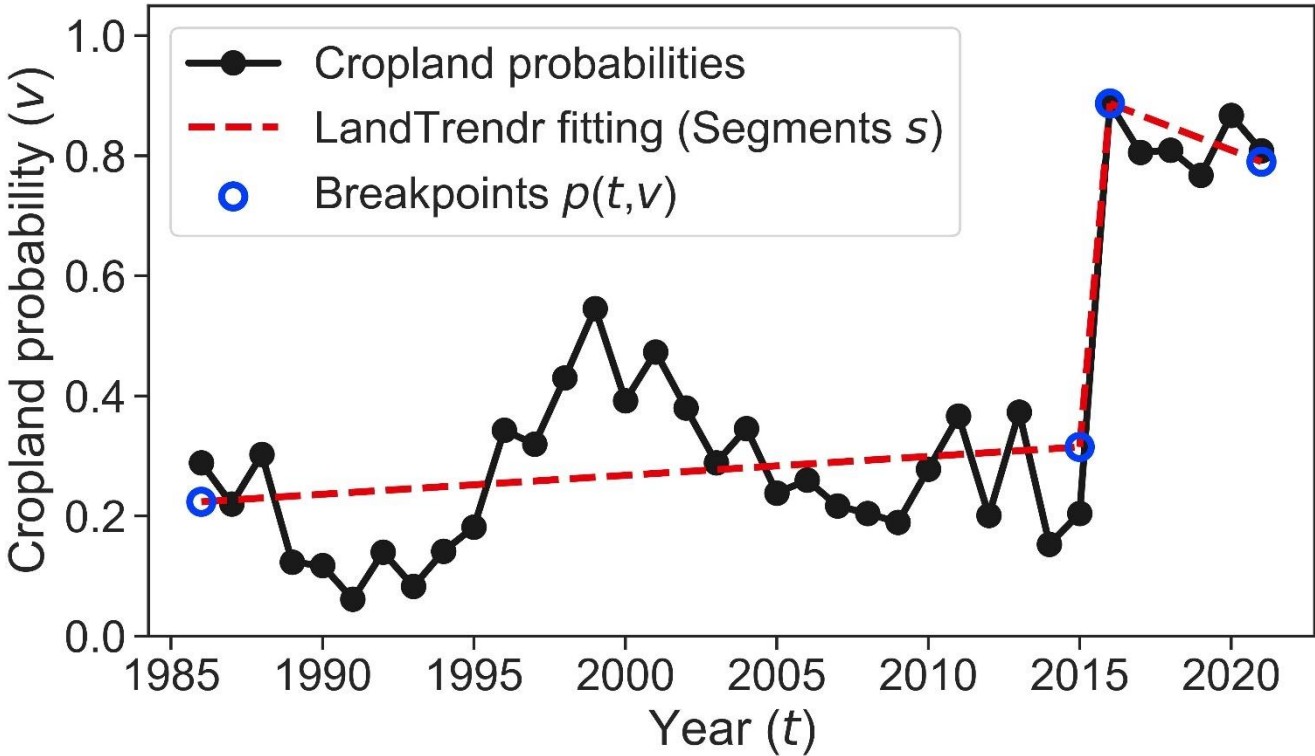

**Figure 2: Schematic diagram of the segmentation results of annual cropland probability time series based on LandTrendr.**



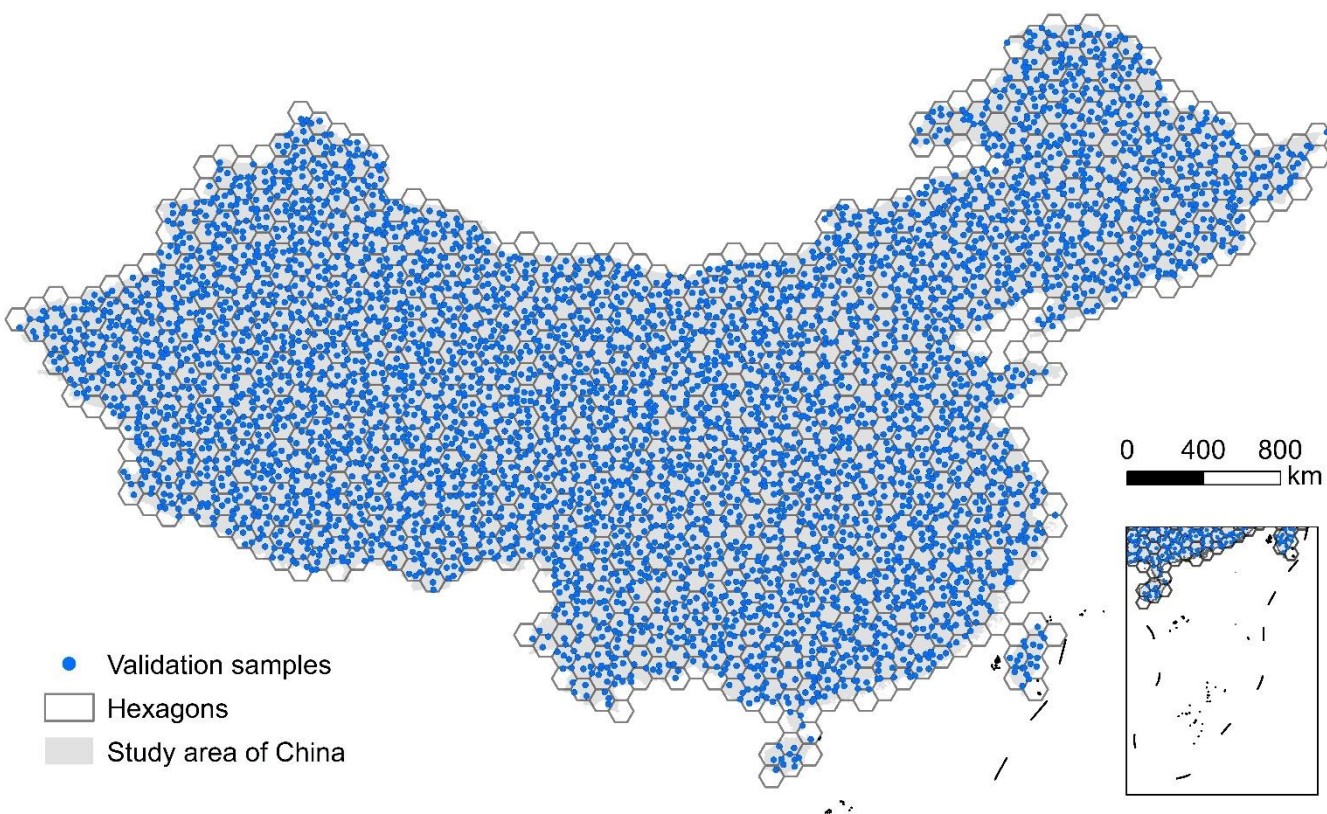


**Figure 3: Distribution of validation samples.**

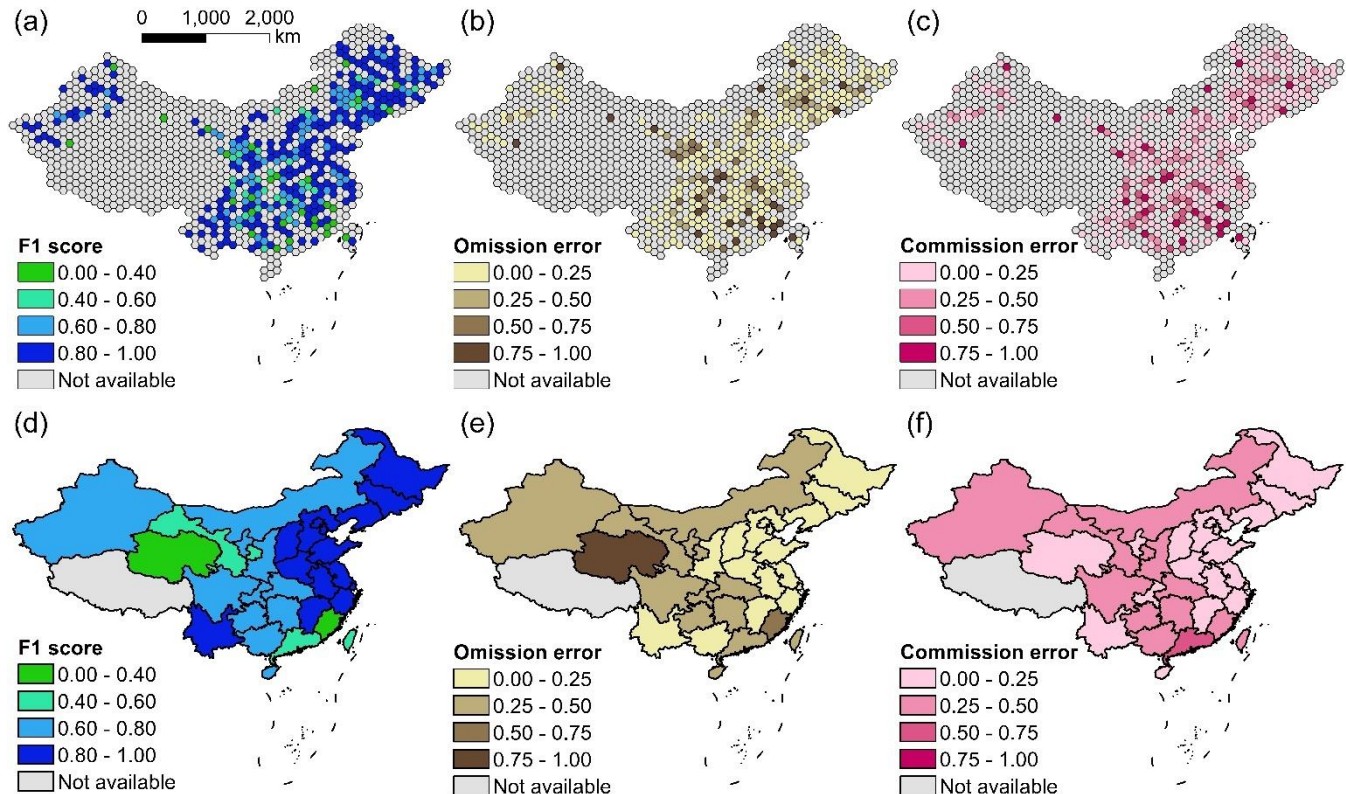

**Figure 4: Spatial variation of mean accuracy of CACD for 1986-2020. (a-c) Hexagon level. Noted only hexagons containing both cropland and non-cropland samples were calculated. (d-f) Province-level. Noted Xizang, Hong Kong, and Macau were excluded from the analysis due to small numbers of cropland samples.**





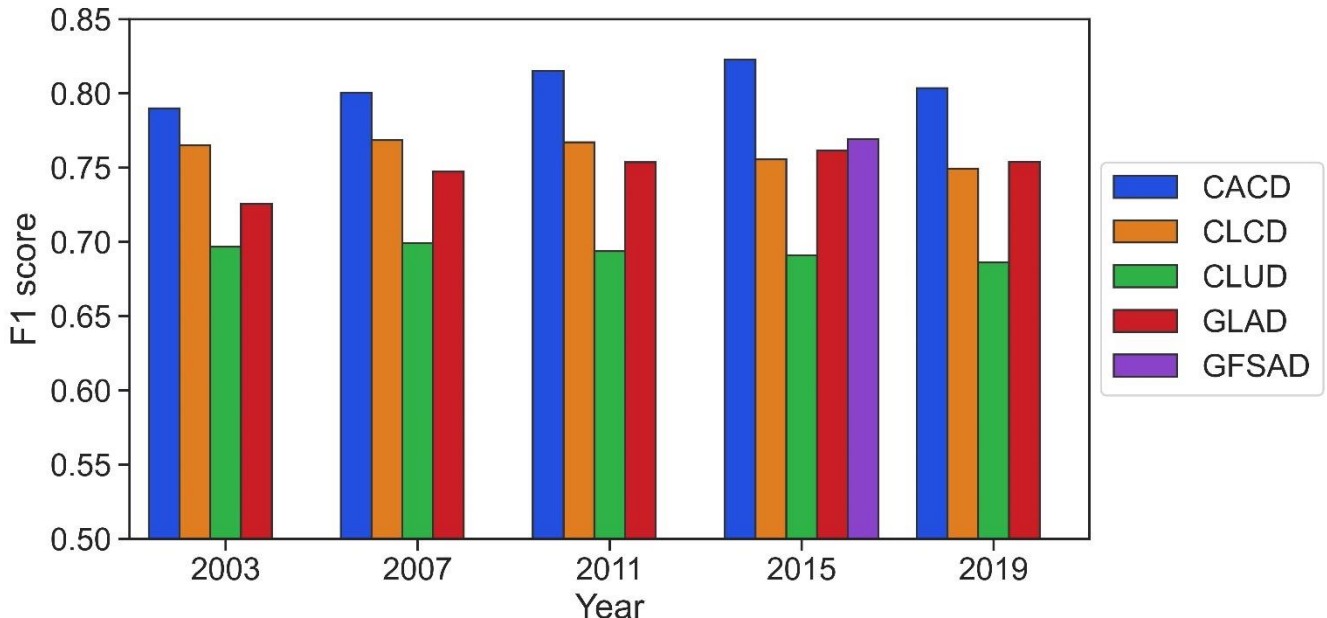

**Figure 5: Comparisons of the pixel-wise accuracy of F1 score between the five products based on the annual validation samples.**

Earth System Discussions
Science
Data

**Figure 6: Spatial agreement of cropland extent of the five products (CACD, CLCD, CLUD, GLAD, and GFSAD) in 2015. Matched level indicates how often a pixel is identified as cropland. For example, '4' means four out of the five products recognize it as cropland. The bar chart in the lower-left corner denotes the area proportion of different matched levels (1-5). Locations A-F respectively correspond to zoomed areas in Figure 7.**



**Figure 7: Regional comparisons of the five products in 2015. Referencing Landsat imagery in the first column is composited by SWIR1, NIR, and Red bands in RGB channels. Columns 2-6 display the spatial distribution of the five products, with cropland shown in white and non-cropland shown in black. All maps have a scale of 1:100000. Locations of A-F are shown in Figure 6.**



**Figure 8: Scatterplots of the provincial cropland area in 2015 between the five products and the statistical data. The red dotted line**
**represents the 1:1 diagonal.**







Figure 9: Cropland area changes in China between 1986-2021. (a) Provincial net loss derived from the CACD data. Negative values indicate cropland gain. (b) Annual dynamics of the map-based and the sample-based cropland area at the national scale.



Figure 10: Examples of cropland expansion and loss between 1986-2021 in four selected regions in China. (a) Chifeng, Inner Mongolia, where large-scale cropland expansion occurred after 2006 for pasture cultivation. (b) Aksu, Xinjiang. The region was mainly dominated by cotton cultivation. (c) Shanghai, where rapid urbanization induced extensive cropland loss. (d) Tongren, Guizhou, where some croplands in the mountainous area had been abandoned due to soil infertile and rural out-migration. All the Landsat images are freely provided by USGS. Both base maps in (c) and (d) are open web maps provided by Esri.



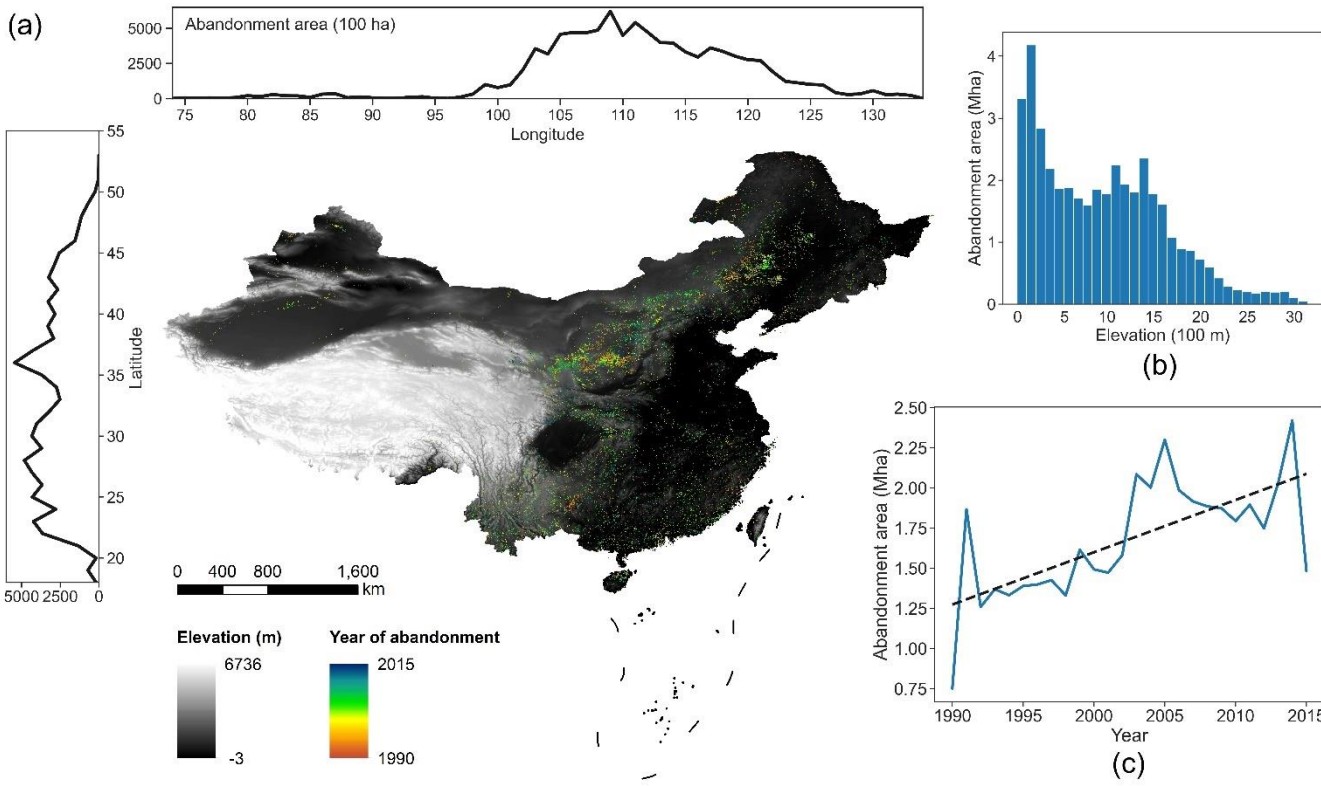


**Figure 11: Cropland abandonment in China between 1990-2015. (a) Spatial distribution of abandonment time. (b) Histogram distribution within different elevation ranges. (c) Annual trends of abandonment area. The elevation information is derived from the SRTM digital elevation dataset (Jarvis et al., 2008).**



## Tables

**Table. 1. Rules for annual cropland transition discrimination based on the LandTrendr segmentation results. $p_{is}(t_{is},v_{is})$ and $p_{ie}(t_{ie},v_{ie})$ are the starting and ending breakpoints of the *i-th* segment $s_i$, respectively, where *t* is the year and *v* is the fitted cropland probability.**

| Fitted cropland probabilities of $p_{is}$ and $p_{ie}$ | Transition type of $s_i$ | Land use type for each year in $s_i$ |
|---|---|---|
| $v_{is}>0.5$, $v_{ie}>0.5$ | Stable cropland | Cropland from $t_{is}$ to $t_{ie}$ |
| $v_{is}\leq0.5$, $v_{ie}\leq0.5$ | Stable non-cropland | Non-cropland from $t_{is}$ to $t_{ie}$ |
| $v_{is}>0.5$, $v_{ie}\leq0.5$ | Cropland loss | Cropland in $t_{is}$ and non-cropland from from $t_{is+1}$ to $t_{ie}$ |
| $v_{is}\leq0.5$, $v_{ie}>0.5$ | Cropland gain | Non-cropland in $t_{is}$ and cropland from from $t_{is+1}$ to $t_{ie}$ |

**Table. 2. Change accuracy of CACD based on the annual validation samples. Change is defined as any cropland use conversion identified between 1986-2021. UA: user's accuracy. PA: producer's accuracy.**

| | | Map | | | |
|---|---|---|---|---|---|
| | | No-change | Change | Total | PA |
| Reference | No-change | 95 | 5 | 100 | 0.95 |
| | Change | 24 | 62 | 86 | 0.72 |
| | Total | 119 | 67 | 186 | |
| | UA | 0.80 | 0.93 | | |
| F1 score: 0.81; Overall accuracy: 0.84; Kappa coefficient: 0.68 | | | | | |

**Table. 3. Comparison of mapping accuracy of CACD, CLUD, CLCD, GLAD, and GFSAD in 2015 based on GLCVSS. F1: F1 score. OA: overall accuracy. Kappa: Kappa coefficient. UA: user's accuracy. PA: producer's accuracy.**

| Product | F1 | OA | UA | PA | Kappa |
|---|---|---|---|---|---|
| CACD | 0.82 | 0.94 | 0.83 | 0.81 | 0.78 |
| CLCD | 0.77 | 0.92 | 0.82 | 0.73 | 0.73 |
| CLUD | 0.70 | 0.90 | 0.72 | 0.67 | 0.64 |
| GLAD | 0.72 | 0.92 | 0.64 | 0.83 | 0.68 |
| GFSAD | 0.76 | 0.93 | 0.77 | 0.76 | 0.72 |

**Table. 4. Comparison of agreement level of CACD, CLUD, CLCD, GLAD, and GFSAD with Geo-Wiki sample set. Noted CACD, CLUD, and CLCD in 2016 are used while GLAD and GFSAD are from 2015.**

| Product | Agreement level |
|---|---|
| CACD | 0.86 |
| CLCD | 0.83 |
| CLUD | 0.65 |
| GLAD | 0.60 |
| GFSAD | 0.73 |