# Peer review of "A 30 m annual cropland dataset of China from 1986 to 2021"

_Earth System Science Data, 2023_

## Author Comment (AC1)

**Comments from Reviewer #1:**

The paper A 30 m annual cropland dataset of China from 1986 to 2021 provides a remarkable attempt at creating national knowledge on the spatial and temporal patterns of cropland in China. In general, it is a well-written and useful study and I enjoy the reading. The following comments are my suggestions for ensuring its messages are clear ad grounded behind the results.

> Thanks for your positive comments. Based on your constructive suggestions, we have carefully revised our manuscript to better present the methods and results. Please find our detailed responses to your comments below.

1. Definition of cropland. Can I say that here you excluded all cash crops, like tea garden, citrus, etc. (in addition to sugarcane), all of which are widely distributed in Southern China. If so, it may be necessary you clearly mentioned this point in your manuscript.

> Thank you for pointing it out. We would like to reiterate the definition of our annual cropland, which is **a piece of land of 0.25 ha in minimum (minimum width of 30 m) that is sowed/planted and harvestable at least once within the 12 months after the sowing or planting date**. One fundamental criterion for discerning annual cropland from other crops is that its vegetation signals in remote sensing imagery must demonstrate noticeable variations over a 12-month period, reflecting the planting and harvesting activities. In this regard, certain exceptions are excluded in the definition of annual cropland: (1) Perennial crops like sugarcane and cassava, which have longer vegetation cycles and are not planted annually. However, if they are planted and harvested within a 12-month timeframe, we would consider them as croplands for that specific year. (2) Fruit, tea, and coffee plantations, as their vegetation signals more closely resemble those of trees. (3) Greenhouse crops, as they exhibit distinct remote sensing characteristics compared to other cropland types. (4) Small plots such as legumes that do not meet the minimum size criteria of cropland.

To further illustrate the differences between our defined annual cropland and other crops (such as tea and citrus that you mentioned), we selected seven agricultural regions, mainly located in Southern China, to compare their remote sensing images and associated NDVI time series. Fig. R1a depicts a typical rice field in Hengyang, Hunan, where the NDVI signals fluctuate periodically within a one-year span. In comparison, other plantations including perennial sugarcane, cassava, fruit trees, tea, coffee, and greenhouses exhibit distinct characteristics (Fig. R1b-g). Given their perennial nature, sugarcane and cassava may not undergo annual harvests, resulting in consistently high NDVI values in certain years (as indicated by the red circles in Fig. R1b-c). Fruit trees, on the other hand, continue to grow over several years rather than being harvested in a single year like conventional crops. Consequently, despite experiencing phenological changes, NDVI time series for fruit trees consistently maintain high values, typically exceeding 0.4 (Fig. R1e-f). Regarding greenhouse crops, their NDVI values are generally lower than those of conventional crops, with the maximum value not surpassing 0.6 (Fig. R1g). Therefore, they are not considered cropland in this study.

[Figure]

**Figure R1**. Comparisons of satellite images and NDVI time series between annual cropland defined in this study and other crops. (a) Rice fields in Hengyang, Hunan. (b) Sugarcane plantation in Zhanjiang, Guangdong. (c) Cassava crops in Nanning, Guangxi. (d) Citrus trees in Ganzhou, Jiangxi. (e) Tea gardens in Wuyishan, Fujian. (f) Coffee trees in Pu'er, Yunnan. (g) Recent greenhouse construction in Ningbo, Zhejiang. All the figures are generated using © Google Earth Engine.

In the revised manuscript Section 2 (Lines 116-127), we have reorganized the definition of cropland for better clarity, which is duplicated as follows:

*"Annual cropland in this study is defined as a piece of land of 0.25 ha in minimum (minimum width of 30 m) that is sowed/planted and harvestable at least once within the*

*12 months after the sowing or planting date. This definition aligns with the criteria established by the Joint Experiment of Crop Assessment and Monitoring (JECAM) network (Defourny et al., 2014) and adopts a shared scope of cropland that meets FAO's Land Cover Meta Language (Di Gregorio, 2005). One crucial criterion for discerning annual cropland in this study is that its vegetation signals in remote sensing imagery must demonstrate noticeable variations over a 12-month period, reflecting the planting and harvesting activities (Fig. S1a). Consequently, certain exceptions are excluded in the definition of annual cropland: (1) Perennial crops like sugarcane and cassava, which have longer vegetation cycles and are not planted annually (Fig. S1b-c). However, if they are planted and harvested within a 12-month timeframe, we would consider them as croplands for that specific year. (2) Fruit, tea, and coffee plantations, as their vegetation signals more closely resemble those of trees (Fig. S1d-f). (3) Greenhouse crops, as they exhibit distinct remote sensing characteristics compared to other cropland types (Fig. S1g). (4) Small plots such as legumes that do not meet the minimum size criteria of cropland."*

Additionally, we have incorporated Fig. R1 into the supplementary materials Fig. S1.

2. Intercomparison. I am happy the CACD was well validated with some published land cover products. However, it seems all selected reference datasets are single/multiple epoch maps. How about the agreement level with some cropland dynamic products, e.g., https://glad.umd.edu/dataset/croplands. In this way we can directly know how good or the accuracy of changed cropland, including both cropland expansion and loss. > Thank you for the valuable suggestion. In the revised manuscript, we have included comparisons of multi-epoch cropland products across regions (Figs. R2-6, respectively corresponding to supplementary materials Figs. S12-16). Taking the GLAD dataset you mentioned as an example, it shows satisfactory performance in northern China (Fig. R2). However, there is a considerable level of misclassification and underestimation of GLAD data in southern China (Figs. R3-5), which is also corroborated in the comparison with statistical data (Fig. 8 in the manuscript). In addition, we identified classification errors in products like CLCD and CLUD, which failed to detect cropland losses due to airport construction in 2017 in Chengdu, Sichuan (Fig. R6). We are confident that these in-depth analyses will provide readers with a more comprehensive and nuanced understanding of the similarities and distinctions among various products.

[Figure]

**Figure R2**. Comparisons of Landsat images and cropland products across years in Altay, Xinjiang, with cropland shown in white and non-cropland shown in black. All the figures are generated using © Google Earth Engine.

[Figure]

**Figure R3**. Comparisons of Landsat images and cropland products across years in Liuzhou, Guangxi, with cropland shown in white and non-cropland shown in black. All the figures are generated using © Google Earth Engine.

[Figure]

**Figure R4**. Comparisons of Landsat images and cropland products across years in Ganzhou, Jiangxi, with cropland shown in white and non-cropland shown in black. All the figures are generated using © Google Earth Engine.

[Figure]

**Figure R5**. Comparisons of Landsat images and cropland products across years in Longyan, Fujian, with cropland shown in white and non-cropland shown in black. All the figures are generated using © Google Earth Engine.

[Figure]

**Figure R6**. Comparisons of Landsat images and cropland products across years in Chengdu, Sichuan, with cropland shown in white and non-cropland shown in black. All the figures are generated using © Google Earth Engine.

---

## Author Comment (AC2)

**Comments from Reviewer #2:**

General comments:

Long-term and accurate cropland monitoring is quite important for provisioning food security and environmental sustainability. This study developed an annual cropland dataset in China (CACD) from 1986 to 2021 by using a novel cost-effective annual cropland mapping framework that integrated time-series Landsat imagery. The authors have done a good job in training and validation dataset selection and annual cropland mapping. The accuracy assessment indicates that CACD has relatively high reliability. Comparisons between CACD and other cropland datasets show its improvements spatially. Overall, I think the CACD is a good annual cropland extent dataset with fine resolution. However, I still have some concerns about the methods and results analysis and have been provided in the specific comments.

> Thanks for your detailed assessment and positive comments. Based on your constructive suggestions, we have carefully revised our manuscript to better present the methods, results, and discussion. Please find our point-by-point responses to your specific comments below.

Specific comments:

1. Lines 61-64. You listed two crop type data (i.e., NASS-CDL, European Union 10 m crop type map) and introduced the research gap, but your dataset also does not include the crop type information and making readers a little disappointed. Meanwhile, I can't agree with "To date, no fine resolution annual cropland dataset of China exists yet". In your literature review, Yang and Huang (2021) developed the 30 m annual land cover dataset in China (CLUD) from 1990 to 2019. There are no essential differences between cropland in this study and cropland from CLUD, because your dataset also doesn't include the crop type information.

> Thank you for your comment. We acknowledge the significance and complexity of crop type mapping, and we are actively engaged in addressing this issue, as highlighted in the Discussion section of the revised manuscript (Lines 415-417). To achieve it, however, one pre-request is to precisely delineate the spatial-temporal distributions of cropland, which constitutes the primary focus of our research. While the CLCD datasets offered by Yang and Huang (2021) did encompass cropland, our assessment indicated that its cropland classification accuracy was comparatively lower than ours (Fig. 5 in the manuscript). Additionally, we observed an overestimation of cropland area in the CLCD dataset (Fig. 8). To bolster the credibility of our findings, especially in capturing dynamic cropland changes, we showed multi-year Landsat images and corresponding cropland classification maps of different datasets in Chengdu, Sichuan Province (Fig. R1). This region, once a traditional agricultural area until 2016, experienced significant losses of croplands for airport construction. Our datasets distinctly identified these transformations, whereas both CLCD and CLUD datasets fell short. This is a simple illustrative example, yet we have found many similar cases during evaluation (see supplementary materials Figs. S12-16). Several key factors contributed to these disparities. On the one hand, our methodologies were specifically tailored for annual cropland mapping, whereas products like CLCD and CLUD encompass a broader

spectrum of land cover types. In their results, croplands were sometimes misclassified as other vegetative cover types. On the other hand, the definition of cropland varied among products. In summary, precise knowledge of where and when croplands are distributed holds paramount importance for subsequent applications like crop type classification. Although existing land cover products may partially address this, they still present limitations and challenges. Therefore, thematic mapping, as exemplified in our study and as many studies on forests (Hansen et al., 2013), water body (Pekel et al., 2016), impervious surfaces (Gong et al., 2020) have been done, remains imperative to enrich our understanding of the Earth's landscape.

[Figure]

**Figure R1**. Comparisons of Landsat images and cropland products across years in Chengdu, Sichuan, with cropland shown in white and non-cropland shown in black. All the figures are generated using © Google Earth Engine.

**References**

Gong, P., Li, X., Wang, J., Bai, Y., Chen, B., Hu, T., Liu, X., Xu, B., Yang, J., Zhang, W., and Zhou, Y.: Annual maps of global artificial impervious area (GAIA) between 1985 and 2018, *Remote Sensing of Environment*, 236, 111510, 2020.

Hansen, M. C., Potapov, P. V., Moore, R., Hancher, M., Turubanova, S. A., Tyukavina, A., Thau, D., Stehman, S. V., Goetz, S. J., Loveland, T. R., Kommareddy, A., Egorov, A., Chini, L., Justice, C. O., and Townshend, J. R. G.: High-Resolution Global Maps of 21st Century Forest Cover Change, *Science*, 342, 850-853, 2013.

Pekel, J.-F., Cottam, A., Gorelick, N., and Belward, A. S.: High-resolution mapping of global surface water and its long-term changes, *Nature*, 540, 418-422, 2016.

2. Lines 103. "The aim of this study is to propose a novel paradigm for large-scale fine-resolution cropland dynamics monitoring." I think the paradigm is not very innovative. A study titled "Forest management in southern China generates short term extensive carbon sequestration" applied a similar framework to analyze the forest dynamics. You two used the same methods: RF-based probability prediction of cropland or forest, and LandTrendr-based segmentation.

> We generously appreciate your insight into the innovation within our methodology. Our core advancement lies in the seamless integration of advanced techniques, including automatic training sample generation, RF-based probability prediction, LandTrendr-based segmentation, rule-based annual cropland discrimination, and post-classification. While we did not develop these methods individually, their collective application, attuned to the specific traits of agricultural landscape in China, stands as a pivotal innovation. Furthermore, we conducted thorough experiments in study area division, sample design, and parameter testing. These seemingly small yet highly nuanced aspects have played a decisive role in the triumph of our novel paradigm for mapping annual cropland dynamics. We firmly believe that our results and insights will significantly contribute to future mapping endeavors in similar domains. By emphasizing these unique elements, we thus convey the innovative aspects of our study and distinguish it from previous work such as forest dynamic analysis. In the Discussion section of the revised manuscript (Lines 378-396), we have reinforced our contributions to the literature, which can be summarized as follows:

*"This research introduces an integrated framework for monitoring annual cropland dynamics at the 30 m spatial resolution. Our methods offer several contributions to the existing literature. First, we leverage baseline land cover maps and the TWDTW discrimination algorithm to realize automated training sample generation. This eliminates the time-consuming and labor-costive process of traditional training sample collection, enabling cost-effective cropland mapping at large scales. Second, we adopt the random forest classifier for annual cropland probabilities estimation and then integrate time series of these probabilities as spectral metrics into LandTrendr for cropland trajectory modeling. The incorporation of machine learning and change detection techniques not only increases accuracy but also improves the spatial-temporal consistency of classification results. Third, we establish a set of transition rules to convert the LandTrendr fitting results to pixel-wise annual cropland types, which can capture various cropland use changes such as abandonment or fallow. This novel strategy is distinct from existing initiatives (Potapov et al., 2022; Dara et al., 2018; Xu et al., 2018; Yin et al., 2020). Our results highlight the potential of change*

*detection algorithms like LandTrendr to complement traditional classification processes in identifying dynamic land cover changes effectively.*

*In practice, we partitioned the study area into nine agricultural zones and performed localized annual cropland mapping within each 0.8°×0.8° subregion. We also evaluated the impacts of training sample size and LandTrendr parameter settings on classification accuracy. Localized classifications may greatly improve the accuracy in heterogeneous regions such as southern China. Our experiments provide valuable insights for future land cover and land use mapping endeavors. Moreover, we took advantage of the powerful data storage, computing, and analysis capabilities of GEE to build an end-to-end framework, which enables fast annual cropland mapping in any given area of interest worldwide. Theoretically, the proposed framework is highly adaptable and can be extended to map other land use types."*

3. Lines 116-117. "Cropland in this study is defined as a piece of land of 0.09 ha in minimum (minimum width of 30 m) that is sowed/planted and harvestable at least once within the 12 months after the sowing or planting date." The definition of cropland in this study differs from that in previous studies. The vegetation indices (e.g., NDVI, EVI) of cropland samplings in the training and validation dataset could reflect the planting or harvest signals. Thus, statistics of vegetation indices variations during the growth period of the samples could improve the reliability rather than depending on visual interpretation only. Additionally, how do you exclude the sugarcane plantation and cassava crop in the training and validation samples? What's the difference of spectral signals between sugarcane plantation/cassava crop and other crops?

> Thanks for pointing out these issues. We would like to reiterate the definition of our annual cropland, which is defined as **a piece of land of 0.25 ha in minimum (minimum width of 30 m) that is sowed/planted and harvestable at least once within the 12 months after the sowing or planting date**. This definition is established based on the widely accepted criteria set forth by the Joint Experiment of Crop Assessment and Monitoring (JECAM) (Defourny et al., 2014) and is consistent with the Food and Agriculture Organization's (FAO) Land Cover Meta Language for cropland (Di Gregorio, 2005). The overarching objective of JECAM is to achieve consensus on methodologies, establish monitoring and reporting protocols, and promote best practices for a diverse range of global agricultural systems (Waldner et al., 2016). Over the past few years, this definition has been applied in various mapping endeavors, particularly in annual cropland mapping (Matton et al., 2015; Jolivot et al., 2021; Zhang and Wu, 2022). We believe that employing this definition enables the global agricultural monitoring community to compare results based on disparate sources of data, using various methods, across a variety of global cropping systems.

As you rightly emphasized, the temporal variations in vegetation indices throughout the growth cycle serve as crucial indicators for cropland identification, and we do take this aspect into consideration during the sample collection process. Our developed Cropland Inspector Tool not only provides visual access to Landsat images across different periods but also presents the time-series NDVI trends for a given area of interest. Within

this framework, a pivotal criterion for discerning annual cropland is that its vegetation signals must demonstrate noticeable variations over a 12-month span, mirroring the planting and harvesting activities. Samples failing to meet this criterion are not categorized as cropland in our research.

Regarding your raised concern about sugarcane plantations and cassava crops, we have included several examples in Fig. R2. It is evident that while the signals of sugarcane and cassava also exhibit periodic changes, they do not consistently demonstrate obvious planting and harvesting variations within a year (red circles in Fig. R2b-c), as is typical for general crops. This is attributed to their perennial nature. Therefore, in such cases, we do not classify them as cropland.

Taking together, we have highlighted our definition of annual cropland in the revised manuscript Section 2 (Lines 116-127) and incorporated Fig. R2 to the supplementary materials Fig. S1. It now stands as follows:

*"Annual cropland in this study is defined as a piece of land of 0.25 ha in minimum (minimum width of 30 m) that is sowed/planted and harvestable at least once within the 12 months after the sowing or planting date. This definition aligns with the criteria established by the Joint Experiment of Crop Assessment and Monitoring (JECAM) network (Defourny et al., 2014) and adopts a shared scope of cropland that meets FAO's Land Cover Meta Language (Di Gregorio, 2005). One crucial criterion for discerning annual cropland in this study is that its vegetation signals in remote sensing imagery must demonstrate noticeable variations over a 12-month period, reflecting the planting and harvesting activities (Fig. S1a). Consequently, certain exceptions are excluded in the definition of annual cropland: (1) Perennial crops like sugarcane and cassava, which have longer vegetation cycles and are not planted annually (Fig. S1b-c). However, if they are planted and harvested within a 12-month timeframe, we would consider them as croplands for that specific year. (2) Fruit, tea, and coffee plantations, as their vegetation signals more closely resemble those of trees (Fig. S1d-f). (3) Greenhouse crops, as they exhibit distinct remote sensing characteristics compared to other cropland types (Fig. S1g). (4) Small plots such as legumes that do not meet the minimum size criteria of cropland."*

[Figure]

**Figure R2**. Comparisons of satellite images and NDVI time series between annual cropland defined in this study and other crops. (a) Rice fields in Hengyang, Hunan. (b) Sugarcane plantation in Zhanjiang, Guangdong. (c) Cassava crops in Nanning, Guangxi. (d) Citrus trees in Ganzhou, Jiangxi. (e) Tea gardens in Wuyishan, Fujian. (f) Coffee trees in Pu'er, Yunnan. (g) Recent greenhouse construction in Ningbo, Zhejiang. All the figures are generated using © Google Earth Engine.

**References**

Defourny, P., Jarvis, I., and Blaes, X.: JECAM Guidelines for cropland and crop type definition and field data collection, JECAM, 2014.

Di Gregorio, A.: Land cover classification system: classification concepts and user manual: LCCS, Food & Agriculture Org.2005.

Jolivot, A., Lebourgeois, V., Leroux, L., Ameline, M., Andriamanga, V., Bellón, B., Castets, M., Crespin-Boucaud, A., Defourny, P., Diaz, S., Dieye, M., Dupuy, S., Ferraz, R., Gaetano, R., Gely, M., Jahel, C., Kabore, B., Lelong, C., le Maire, G., Lo Seen, D., Muthoni, M., Ndao, B., Newby, T., de Oliveira Santos, C. L. M., Rasoamalala, E., Simoes, M., Thiaw, I., Timmermans, A., Tran, A., and Bégué, A.: Harmonized in situ datasets for agricultural land use mapping and monitoring in tropical countries, *Earth Syst. Sci. Data*, 13, 5951-5967, 10.5194/essd-13-5951-2021, 2021.

Matton, N., Canto, G. S., Waldner, F., Valero, S., Morin, D., Inglada, J., Arias, M., Bontemps, S., Koetz, B., and Defourny, P.: An Automated Method for Annual Cropland Mapping along the Season for Various Globally-Distributed Agrosystems Using High Spatial and Temporal Resolution Time Series, *Remote Sensing*, 10.3390/rs71013208, 2015.

Waldner, F., De Abelleyra, D., Verón, S. R., Zhang, M., Wu, B., Plotnikov, D., Bartalev, S., Lavreniuk, M., Skakun, S., Kussul, N., Le Maire, G., Dupuy, S., Jarvis, I., and Defourny, P.: Towards a set of agrosystem-specific cropland mapping methods to address the global cropland diversity, *International Journal of Remote Sensing*, 37, 3196-3231, 10.1080/01431161.2016.1194545, 2016.

Zhang, M. and Wu, B.: Global 30-m spatial distribution of cropland in 2020 (GCL30_2020) [dataset], 10.12237/casearth.62ff4caa819aec75a535cbe6, 2022.

4. Lines 146-147. As you said, "The threshold value was set following recommendations by Ghorbanian et al. (2020)". But I didn't find a threshold table to show the difference among the nine agricultural zones. In each subregion, ~800 training samples were used. So, how many cropland and non-cropland samples are there in each subregion?

> Thank you for pointing out this issue. The threshold mentioned here corresponds to 20% following Ghorbanian et al. (2020). In practical terms, we generated a pool of 4000 potential samples from the intersected area of CLCD and CLUD datasets for each 0.8°×0.8° subregion. From this pool, we selected and retained 800 samples (20%) with the lowest dissimilarity value compared to the referencing set. This underscores our use of an areal-proportional allocation-based sampling strategy - a common method (Huang et al., 2002; Jin et al., 2014; Zhang et al., 2023) in land cover mapping initiatives. However, it is worth noting that in the application of areal-proportional sampling, rare land-cover types may have smaller sample sizes and may consequently be underrepresented, leading to their poor classification performance. To mitigate this, Zhu et al. (2016) recommended a minimum sample size of 600 for rare land-cover types. A recent study also used this parameter for sample balancing in global annual land cover mapping within each 5°×5° geographical tile (Zhang et al., 2023). Given that the subregion size in our research (0.8°×0.8°) is considerably smaller than that in Zhang's study (5°×5°), we established the minimum sample size as 100. This means that we initially generated 800 samples for each 0.8°×0.8° subregion including both cropland and non-cropland categories. If either category fell below the specified threshold, we increased it to a minimum of 100 samples.

We have revised the manuscript Section 2.1 (Lines 154-162) for a clearer explanation of our sample design, which is duplicated as follows:

*"In practice, we employed the widely adopted areal-proportional sampling strategy (Huang et al., 2002; Jin et al., 2014) for allocating both cropland and non-cropland samples. However, one limitation of this approach is that it can result in extremely small sample sizes for rare land-cover types in homogeneous landscapes. When these types are underrepresented in the samples, it may lead to subpar classification performance. To address this concern, Zhu et al. (2016) recommended a minimum sample size of 600 for rare land-cover types. A recent study also applied this parameter for sample balancing in global annual land cover mapping within each 5°×5° geographical tile (Zhang et al., 2023). Given that the subregion size in our research (0.8°×0.8°) is significantly smaller than that in Zhang's study (5°×5°), we established the minimum sample size as 100. This means that we initially generated 800 samples for each 0.8°×0.8° subregion, encompassing both cropland and non-cropland categories. If either category fell below the specified threshold, we increased it to a minimum of 100 samples."*

**References**

Huang, C., Davis, L. S., and Townshend, J. R. G.: An assessment of support vector machines for land cover classification, *International Journal of Remote Sensing*, 23, 725-749, 10.1080/01431160110040323, 2002.

Jin, H., Stehman, S. V., and Mountrakis, G.: Assessing the impact of training sample selection on accuracy of an urban classification: a case study in Denver, Colorado, *International Journal of Remote Sensing*, 35, 2067-2081, 2014.

Zhang, X., Zhao, T., Xu, H., Liu, W., Wang, J., Chen, X., and Liu, L.: GLC_FCS30D: The first global 30-m land-cover dynamic monitoring product with a fine classification system from 1985 to 2022 using dense time-series Landsat imagery and continuous change-detection method, *Earth Syst. Sci. Data Discuss*., 2023.

Zhu, Z., Gallant, A. L., Woodcock, C. E., Pengra, B., Olofsson, P., Loveland, T. R., Jin, S., Dahal, D., Yang, L., and Auch, R. F.: Optimizing selection of training and auxiliary data for operational land cover classification for the LCMAP initiative, *ISPRS Journal of Photogrammetry and Remote Sensing*, 122, 206-221, 2016.

5. Lines 176-207. I think these two steps are important for the final cropland layer. The authors give two examples (Figure 2 and Figure S2) to illustrate how the LandTrendr algorithm works. I think more examples should be given to prove the robustness of the cropland mapping method. For example, how cropland probabilities and vegetation indices changed when cropland was converted to urban/grassland/forest, and grassland/forest was reclaimed to cropland.

> Thank you for your valuable suggestion. We have selected four regions, each representing a unique agricultural landscape, to exemplify dynamic cropland changes in CACD (Figs. R3-6). Specifically, these four regions denote scenarios of cropland to

urban land, cropland to forest, grass to cropland, and forest to cropland, respectively. In these cases, subfigure (a) displays the original Landsat images alongside corresponding high-resolution images sourced from Google Earth for selected regions spanning multiple years. Subfigure (b) presents the corresponding NDVI time series derived from all available Landsat data. Subfigure (c) provides comprehensive details, including the estimated cropland probabilities, LandTrendr segmentation results, and the final cropland mapping outcomes for the specific point of interest. These visualizations effectively demonstrate the accuracy and robustness of our proposed methodologies in discerning dynamic cropland changes across diverse agricultural landscapes. We have attached these examples and analyses to the supplementary materials Figs. S6-9, which we believe significantly enhance the comprehensiveness of our study.

[Figure]

**Figure R3**. An illustration of cropland to urban land conversion in Chengdu, Sichuan. (a) Landsat and © Google Earth high-resolution images over time. (b) NDVI time series for the selected point of interest. (c) Estimated cropland probabilities, LandTrendr segmentations, and final mapping outcomes (green: cropland, white: non-cropland) for the selected point. All the Landsat images are freely provided by USGS.

[Figure]

**Figure R4**. An illustration of cropland to forest conversion in Yan'an, Shaanxi. (a) Landsat and © Google Earth high-resolution images over time. (b) NDVI time series for the selected point of interest. (c) Estimated cropland probabilities, LandTrendr segmentations, and final mapping outcomes (green: cropland, white: non-cropland) for the selected point. All the Landsat images are freely provided by USGS.

[Figure]

**Figure R5**. An illustration of grass to cropland conversion in Chifeng, Inner Mongolia. (a) Landsat and © Google Earth high-resolution images over time. (b) NDVI time series for the selected point of interest. (c) Estimated cropland probabilities, LandTrendr segmentations, and final mapping outcomes (green: cropland, white: non-cropland) for the selected point. All the Landsat images are freely provided by USGS.

[Figure]

**Figure R6**. An illustration of forest to cropland conversion in Shaoguan, Guangdong. (a) Landsat and © Google Earth high-resolution images over time. (b) NDVI time series for the selected point of interest. (c) Estimated cropland probabilities, LandTrendr segmentations, and final mapping outcomes (green: cropland, white: non-cropland) for the selected point. All the Landsat images are freely provided by USGS.

6. Lines 217-218. A spatial-temporal consistency check approach proposed by Li et al. (2015) was applied to refine the annual cropland maps. I don't think this consistency check algorithm can be used to cropland without any improvements. In Li et al. (2015), there is a very important assumption that "…the transition from urban to other land cover types is not likely and should be avoided… (Section 2.3.2 in Li et al. (2015))". However, the conversion rule of cropland differs from urban land. More descriptions should be given if there are any improvements to this algorithm.

> Thank you for bringing this to our attention. In Li's study, the temporal consistency check approach consists of two major components: temporal filtering and logic reasoning. The strong assumption you mentioned was mainly considered in the latter one ("…the obtained sequence may not be logical, containing urban and non-urban

segments occurring alternatively. Therefore, the logical reasoning check was applied."). However, our spatial-temporal consistency check approach was basically followed and modified from the temporal filtering method. Specifically, for each pixel $i$ in year $t$, we calculated its spatial-temporal consistency probability $Prob_{i,t}$ within the surrounding 3×3×3 window:

$$Prob_{i,t} = \frac{1}{N} \sum_{t'=t-1}^{t+1} \sum_{x=m-1}^{m+1} \sum_{y=n-1}^{n+1} Con(L_{i,t} = L_j), \qquad (R1)$$

where $L_{i,t}$ denotes the label of the target pixel $i$ in year $t$, and $L_j$ denotes the label of pixels in the neighborhood window. $N$ signifies the total number of pixels (i.e., N=27), and $x$ and $y$ indicate the coordinates of pixel $i$ within the window. The core of this approach is the consistency check function $Con()$, which equals 1 if $L_{i,t} = L_j$, and 0 otherwise. Here we employed the threshold of 0.5, as suggested by Li et al. (2015), to discern the label transition between cropland (1) and non-cropland (0): if $Prob_{i,t}$ is less than 0.5, then the label of pixel $i$ in year $t$ is altered to the opposite category.

In the revised manuscript, we have incorporated the above information into Section 2.6 (Lines 231-239) for improved clarity.

7. Lines 264-265. Why do western and southeastern coastal areas have relatively low accuracy (F1 score)? Some explanations should be given. Is it because the cropland in southeastern coastal areas more fragmented?
> Thank you for proposing the point. The fragmentation of cropland does pose a huge challenge for accurate classification, particularly in the western and southeastern coastal areas. Recent studies have underscored notable discrepancies between products in these regions (Lu et al., 2016; Zhang et al., 2022; Xue et al., 2023). According to their results, factors such as intricate land use patterns, smallholder farming practices, and unique topographical features contribute to lower agreement levels. Therefore, it is imperative to conduct comprehensive investigations in these specific areas to improve mapping precision. We have emphasized this in the Discussion section of the revised manuscript (Lines 413-426), which is reiterated below:

*"A few limitations regarding methodology and data are included in the study. First, we track cropland dynamics on an annual basis but do not account for intra-annual variations in crops, such as different crop calendars and cropping intensity. Consequently, our results may not be effective in identifying perennial crops. Second, we depict the general extent of croplands but do not differentiate specific types. This becomes essential when evaluating yields or climate responses of different crops. Future work needs to explore advanced methods for fine-resolution crop types mapping. Third, despite the high accuracy, CACD is subject to several levels of uncertainty. Temporally, CACD has a relatively low accuracy before 2000, given the uneven coverage of Landsat 5 data. Spatially, mapping accuracy varied across different regions, where the western and southeastern coastal areas have a comparatively high error compared to others. Accurately classifying cropland in these regions has consistently been challenging, with recent research highlighting significant*

*discrepancies between products (Lu et al., 2016; Zhang et al., 2022b; Xue et al., 2023). This challenge stems from characteristics of local topography and landscapes, including factors like elevation, slope, field size, and farmland fragmentation (Lu et al., 2016; Zhang et al., 2022b; Xue et al., 2023). Besides, the low quality of Landsat data (e.g., unmasked clouds), as well as the sensitivity of LandTrendr parameters, further constitute the influence on mapping outcomes. Looking ahead, it is crucial to embark on comprehensive investigations and formulate targeted adaptation strategies to enhance the accuracy of cropland classification in these specific areas."*

**References**

Lu, M., Wu, W., Zhang, L., Liao, A., Peng, S., and Tang, H.: A comparative analysis of five global cropland datasets in China, *Science China Earth Sciences*, 59, 2307-2317, 10.1007/s11430-016-5327-3, 2016.

Xue, J., Zhang, X.-l., Chen, S.-c., Hu, B.-f., Wang, N., and Shi, Z.: Quantifying the agreement and accuracy characteristics of four satellite-based LULC products for cropland classification in China, *Journal of Integrative Agriculture*, https://doi.org/10.1016/j.jia.2023.06.005, 2023.

Zhang, C., Dong, J., and Ge, Q.: Quantifying the accuracies of six 30-m cropland datasets over China: A comparison and evaluation analysis, *Computers and Electronics in Agriculture*, 197, 106946, https://doi.org/10.1016/j.compag.2022.106946, 2022.

8. Line 316. "Additionally, cropland areas in some inland provinces (such as Guizhou) remained rather stable." The area of Guizhou province should be rechecked. As I know, Guizhou is the core area of ecological restoration projects of the karst region. Cropland was converted into forest (Yue et al., 2020, Landscape Ecology).

> Thank you for your suggestion. We have acquired statistics on provincial farmland area from the Guizhou Statistical Yearbook, covering years as those mentioned in the reference paper (i.e., 2005-2016). We then compared the data with the cropland area calculated from the five products (Fig. R7). Our analysis reveals that neither the statistical data nor the remote sensing products show significant fluctuations in cropland area in Guizhou Province. Therefore, our initial discussion remains unbiased. While the issue of ecological restoration in the Karst region that you mentioned is indeed valid, it's worth noting that this perspective focuses solely on the loss of cropland, without accounting for its expansion. Additionally, differences in the definition of cropland and other relevant factors may have contributed to the misunderstanding in this context. We greatly appreciate your attention to this matter and value the opportunity to clarify it.

[Figure]

**Figure R7**. Cropland area comparison of Guizhou province among the five cropland products and official statistics.

9. Lines 328-330. "In the Ar Horqin Banner of Chifeng city, Inner Mongolia, large-scale croplands were developed for pasture reclamation and cultivation during the past decades". It should be noted that pasture is a type of grassland rather than crops.

> Thank you for your observation. We re-examined the site and confirmed that the dominant crop type was indeed alfalfa, an important forage source for grassland agricultural development. As a perennial legume, alfalfa is extensively cultivated in Northern China for its high-quality forage, serving as crucial feed for livestock, especially dairy cattle and horses (Wang et al., 2022; Du et al., 2023). As shown in Fig. R5, remote sensing signals of alfalfa are notably distinct from those of pasture. In the revised manuscript Section 4.1 (Lines 350-351), we have rectified it as follows:

*"In the Ar Horqin Banner of Chifeng city, Inner Mongolia, large-scale croplands were developed for the reclamation and cultivation of crops over the past few decades."*

**References**

Du, G., Wang, X., Wang, J., Liu, Y., and Zhang, H.: Analysis of the Spatial–Temporal Pattern of the Newly Increased Cultivated Land and Its Vulnerability in Northeast China, *Land*, 12, 10.3390/land12040796, 2023.

Wang, R., Shi, F., and Xu, D.: The Extraction Method of Alfalfa (Medicago sativa L.) Mapping Using Different Remote Sensing Data Sources Based on Vegetation Growth Properties, *Land*, 11, 10.3390/land11111996, 2022.

"Similarly, vast agricultural land parcels sprang up in Aksu, Xinjiang for cotton cultivation." The newly developed dataset doesn't include crop type information, how do you get this conclusion? Some studies about cotton expansion in Xinjiang Province should be cited to support your conclusion.

> Thank you for your suggestion. We have included two references (Li et al., 2021; Liu, 2022) to substantiate our statement.

**References**

Li, Q., Liu, G., and Chen, W.: Toward a Simple and Generic Approach for Identifying Multi-Year Cotton Cropping Patterns Using Landsat and Sentinel-2 Time Series, *Remote Sensing*, 13, 10.3390/rs13245183, 2021.

Liu, G.: Understanding cotton cultivation dynamics in Aksu Oases (NW China) by reconstructing change trajectories using multi-temporal Landsat and Sentinel-2 data, *Geocarto International*, 37, 4406-4424, 2022.

10. Lines 336-354. In this part, the authors give much information about cropland abandonment in China. The newly developed shows the cropland loss in the Loess Plateau and Beijing–Tianjin Sand Source Control Project zone (Figure 11). However, there is only a little analysis about the cause of cropland abandonment or cropland loss. For example, cropland loss is mainly driven by the "Grain for Green" ecological project in Shanxi and Inner Mongolia. Cropland abandonment is also affected by factors such as lack of labor and low income (Zhang et al., 2019, Acta Geography Sinica).

> We appreciate your diligent review. In the past, due to limited high-resolution data and efficient algorithms, detecting and monitoring cropland abandonment remained challenging and was often conducted at a regional scale (Dara et al., 2018; Yin et al., 2018; Yin et al., 2020). With the advent of CACD, characterized by its high spatial-temporal resolution and comprehensive coverage, we now possess a powerful tool to analyze the spatial-temporal changes of cropland abandonment in China. Our results revealed that approximately 419,342 km$^2$ (17.57%) of croplands were abandoned during 1990-2015, with the central and western regions experiencing the most significant abandonment. These findings are in consonance with those of Li et al. (2018), who estimated provincial cropland abandonment areas based on statistical data. This serves as a compelling example of utilizing CACD in various applications. Going further, we cannot only investigate the driving forces behind China's cropland abandonment but also assess the impacts of abandonment on ecosystem service and biodiversity, all of which have garnered significant interest in the scientific community (Daskalova and Kamp, 2023; Crawford Christopher et al.). Given that this paper is a **data description article**, our primary focus lies in data production and validation. While we acknowledge the importance of understanding the cause of cropland abandonment or cropland loss, more in-depth analyses are expected to be conducted in the future. We have emphasized this in the Discussion section of the revised manuscript (Lines 397-411), which is duplicated as follows:

*"Based on the aforementioned mapping scheme, we produced the first 30-m annual cropland maps of China for the period 1986-2021. The accuracy of the dataset was validated through three independent validation sample sets and multi-perspective comparisons with other products. The CACD dataset is a valuable contribution to the field of agriculture and land use management in China. Its accuracy and high-*

*resolution information provide insights into the changing dynamics of cropland use over time, helping inform policymakers and stakeholders in developing sustainable land use practices. Our subsequent analyses with CACD revealed a dramatic but heterogeneous change in cropland dynamics in China, with over a third of croplands experiencing at least one change of land use during the study period. Furthermore, we identified a total of 419,342 km$^2$ (17.57%) of abandoned croplands from 1990 to 2015, mostly located in the central and western mountainous areas. With the ability to detect cropland abandonment patterns, targeted interventions can be implemented to bring abandoned land back into use, promoting food security and protecting arable land resources. Further efforts can also be directed towards scrutinizing the underlying driving forces of cropland abandonment as well as assessing its impacts on ecosystem service and biodiversity. Additionally, CACD can be utilized to conduct agriculture-related studies, such as analyzing crop yield and productivity, assessing the impact of climate change on crop growth, and monitoring land use changes over time. Overall, CACD represents a powerful tool for promoting sustainable agriculture practices and ensuring the long-term availability of arable land resources in China."*

**References**

Crawford Christopher, L., Yin, H., Radeloff Volker, C., and Wilcove David, S.: Rural land abandonment is too ephemeral to provide major benefits for biodiversity and climate, *Science Advances*, 8, eabm8999, 2022.

Dara, A., Baumann, M., Kuemmerle, T., Pflugmacher, D., Rabe, A., Griffiths, P., Hölzel, N., Kamp, J., Freitag, M., and Hostert, P.: Mapping the timing of cropland abandonment and recultivation in northern Kazakhstan using annual Landsat time series, *Remote Sensing of Environment*, 213, 49-60, https://doi.org/10.1016/j.rse.2018.05.005, 2018.

Daskalova, G. N. and Kamp, J.: Abandoning land transforms biodiversity, *Science*, 380, 581-583, 10.1126/science.adf1099, 2023.

Li, S., Li, X., Sun, L., Cao, G., Fischer, G., and Tramberend, S.: An estimation of the extent of cropland abandonment in mountainous regions of China, *Land Degradation & Development*, 29, 1327-1342, 2018.

Yin, H., Prishchepov, A. V., Kuemmerle, T., Bleyhl, B., Buchner, J., and Radeloff, V. C.: Mapping agricultural land abandonment from spatial and temporal segmentation of Landsat time series, *Remote Sensing of Environment*, 210, 12-24, https://doi.org/10.1016/j.rse.2018.02.050, 2018.

Yin, H., Brandão, A., Buchner, J., Helmers, D., Iuliano, B. G., Kimambo, N. E., Lewińska, K. E., Razenkova, E., Rizayeva, A., Rogova, N., Spawn, S. A., Xie, Y., and Radeloff, V. C.: Monitoring cropland abandonment with Landsat time series, *Remote Sensing of Environment*, 246, 111873, 2020.

11. Figure 3. The cropland and non-cropland samples could be symbolized with different colors

> Many thanks. We have revised the figure symbology for a better visual distinction.

12. Figure 9. The title of the legend is a little weird. "Loss area" should be "Area change" or "Cropland area change". Additionally, this figure only shows the net change of cropland area. When comparing the total area of cropland gain (increase) and loss during the period, the spatial shift of cropland will be more significant.

> Thank you for your valuable feedback. The legend title of Fig. 9 in the revised manuscript has been updated as "Area change". The CACD dataset has now been published in a free data repository (Tu et al., 2023), allowing users to explore regions of their interest. Furthermore, we have generated an online visualization platform using CACD to show the spatial-temporal changes of cropland in China from 1986 to 2021, which can be accessed through the following link:
https://thutyecology.users.earthengine.app/view/cacd-viewer.

**References**

Tu, Y., Wu, S., Chen, B., Weng, Q., Gong, P., Bai, Y., Yang, J., Yu, L., and Xu, B.: A 30 m annual cropland dataset of China from 1986 to 2021 [dataset], 2023. https://zenodo.org/record/7936885

---

## Author Response (AR2)

**Comments from the Editor:**

The authors have addressed most of the reviewers' comments. Please address the comments from report #2.

> We sincerely thank the editor and reviewers for their precious time and recognition of our work. We have carefully addressed all the concerns raised by Reviewer #2 and made revisions to the manuscript accordingly. Please find our point-by-point responses to his/her specific comments below.

**Comments from Reviewer #2**

Accurate, detailed, and up-to-date information on cropland extent is crucial for provisioning food security and environmental sustainability. This study developed an annual cropland dataset in China (CACD) from 1986 to 2021 at 30 m spatial resolution by using Landsat TM/ETM/OLI images and a cost-effective cropland mapping framework.

The revised manuscript is well-written organized. It includes detail description about the mapping framework and some interesting findings derived from the new dataset. The CACD data has a relatively high accuracy, and also improved a lot spatially by comparing with previous cropland dataset (e.g., CLCD, CLUD). Overall, the CACD data is a good cropland extent dataset with fine resolution and can be applied to quantify the ecological consequences of cropland dynamics. Additionally, the authors have done a great job in response the comments from previous reviewers. A few more areas for improvement include:

> Thank you for your encouraging feedback. Incorporating your constructive suggestions, we have carefully revised our manuscript to enhance the clarity of the methods and discussion sections. Below, we provide detailed responses to each of your comments.

1. The definition of cropland is important in land cover mapping. In the revised manuscript, the author refines the definition of cropland as Joint Experiment of Crop Assessment and Monitoring (JECAM) network and adopts a shared scope of cropland that meets FAO's Land Cover Meta Language. To prove the reliability of the newly developed cropland data, the authors conducted a series of comparisons between five cropland datasets (i.e., CLCD, CLUD, GLAD, GFSAD) and statistical data. However, the definitions of these datasets are different, making it hard to conduct data comparison. I suggest adding a supplementary table of cropland definitions and discuss about it in section 3.2.

> Thank you for the valuable suggestion. We acknowledge that the varying definitions of croplands employed by these datasets may lead to potential bias in cross-product comparison. In response, we have added a supplementary table detailing the definitions used by each dataset (Table R1, corresponding to Table S5 in the supplementary materials).

**Table. R1.** Definitions of cropland in CACD, CLCD, CLUD, GLAD, and GFSAD.

| Product | Cropland definition | Reference |
|---------|---------------------|-----------|
| CACD | Defined as a piece of land of 0.25 ha in minimum (minimum width of 30 m) that is sowed/planted and harvestable at least once within the 12 months after the sowing or planting date. This definition excludes:
• Perennial crops like sugarcane and cassava
• Fruit, tea, and coffee plantations
• Greenhouse crops
• Small plots such as legumes that do not meet the minimum size criteria of cropland | This study |
| CLCD | Defined as cultivated lands for crops. Including: mature cultivated land, newly cultivated land, fallow, shifting cultivated land; intercropping land such as crop-fruiter, crop-mulberry, and crop-forest land in which a crop is a dominant species; bottomland and beach that cultivated for at least 3 years. | (Yang and Huang, 2021) |
| CLUD | Defined as cultivated lands for crops. Including: mature cultivated land, newly cultivated land, fallow, shifting cultivated land; intercropping land such as crop-fruiter, crop-mulberry, and crop-forest land in which a crop is a dominant species; bottomland and beach that cultivated for at least 3 years. | (Xu et al., 2020) |
| GLAD | Defined as land used for annual and perennial herbaceous crops for human consumption, forage (including hay) and biofuel. Perennial woody crops, permanent pastures and shifting cultivation are excluded from the definition. The fallow length is limited to 4 years for the cropland class. | (Potapov et al., 2022) |
| GFSAD | Net cropland extent mapped was defined as the sum of the following agricultural croplands:
• Cropland that is cultivated and harvested for food, feed, and (or) fiber, one or more times during a 12-month period;
• Cropland that is left fallow, even when equipped for agriculture; and
• Cropland that is permanently cropped with plantations (for example, orchards, vineyards, coffee, tea, and rubber).
Notably, pasture land is not part of the cropland, except for alfalfa in the United States and some other countries. | (Thenkabail et al., 2021) |

Noted both CLCD and CLUD were generated based on the China land-use/cover datasets developed by the Chinese Academy of Sciences, which encompassed six primary land cover classes (level-1) and 25 sub-classes (level-2) (Liu et al., 2014; Liu et al., 2005; Liu et al., 2003). Following Yang and Huang (2021) and Xu et al. (2020), we adopted their definition of cropland for the level-1 class.

In Section 2.7 (Lines 279-281), the manuscript has been revised as follows: "*It's noted that while all these products delineate cropland extents at a 30 m spatial resolution, they adopt inconsistent definitions of cropland (as detailed in Table S5). As a consequence, the cross-product results could potentially be biased, and we discussed this uncertainty in the limitations and prospects sections (refer to Section 4.3).*"

Moreover, we have discussed the uncertainty of the differences in cropland definition among products in the Discussion section of the revised manuscript (Lines 429-436), which is duplicated as follows: "*Lastly, it is crucial to acknowledge that the datasets compared in this study utilize diverse definitions of cropland (Table S5), potentially leading to discrepancies in cross-product comparisons. For instance, the cropland definition employed in this study excludes perennial crops, while GLAD includes perennial herbaceous crops. Additionally, CLCD, CLUD, and GFSAD may encompass lands cropped with plantations, such as fruits, coffee, and tea, which are excluded in this study (Fig. S1). Furthermore, GLAD and GFSAD incorporate fallow land (GLAD limits the fallow length to 4 years, as it operates cropland mapping on a four-year interval, whereas GFSAD was only mapped for 2015 and did not specify the duration of fallow). These variations in definitions may result in certain overestimation or underestimation of cropland extent across different products.*"

**References**

Liu, J., Kuang, W., Zhang, Z., Xu, X., Qin, Y., Ning, J., Zhou, W., Zhang, S., Li, R., Yan, C., Wu, S., Shi, X., Jiang, N., Yu, D., Pan, X., & Chi, W. (2014). Spatiotemporal characteristics, patterns, and causes of land-use changes in China since the late 1980s. *Journal of Geographical Sciences, 24*, 195-210

Liu, J., Liu, M., Tian, H., Zhuang, D., Zhang, Z., Zhang, W., Tang, X., & Deng, X. (2005). Spatial and temporal patterns of China's cropland during 1990–2000: An analysis based on Landsat TM data. *Remote Sensing of Environment, 98*, 442-456

Liu, J., Liu, M., Zhuang, D., Zhang, Z., & Deng, X. (2003). Study on spatial pattern of land-use change in China during 1995–2000. *Science in China Series D: Earth Sciences, 46*, 373-384

Potapov, P., Turubanova, S., Hansen, M.C., Tyukavina, A., Zalles, V., Khan, A., Song, X.-P., Pickens, A., Shen, Q., & Cortez, J. (2022). Global maps of cropland extent and change show accelerated cropland expansion in the twenty-first century. *Nature Food, 3*, 19-28

Thenkabail, P.S., Teluguntla, P.G., Xiong, J., Oliphant, A., Congalton, R.G., Ozdogan, M., Gumma, M.K., Tilton, J.C., Giri, C., Milesi, C., Phalke, A., Massey, R., Yadav, K., Sankey, T., Zhong, Y., Aneece, I., & Foley, D. (2021). Global cropland-extent product at 30-m resolution (GCEP30) derived from Landsat satellite time-series data for the year 2015 using multiple machine-learning algorithms on Google Earth Engine cloud. In, *Professional Paper* (p. 63). Reston, VA

Xu, Y., Yu, L., Peng, D., Zhao, J., Cheng, Y., Liu, X., Li, W., Meng, R., Xu, X., & Gong, P. (2020). Annual 30-m land use/land cover maps of China for 1980–2015 from the integration of AVHRR, MODIS and Landsat data using the BFAST algorithm. *Science China Earth Sciences, 63*, 1390-1407

Yang, J., & Huang, X. (2021). The 30 m annual land cover dataset and its dynamics in China from 1990 to 2019. *Earth System Science Data, 13*, 3907-3925

2. The cropland probability was estimated by using random forest model, but the description about how to conduct the model is not very clear. Was the random forest model trained at hexagon-level? If each hexagon has a unique random forest model, clarify it in section 2.3.

> Thanks for pointing out this issue. The random forest model was trained exclusively for each or each 0.8°×0.8° subregion (Fig. R2, corresponding to Fig. S2 in the supplementary materials). We have clarified this in Section 2.3 of the revised manuscript (Lines 187-188). It now reads as follows: "*In practice, we implemented a random forest model for each 0.8°×0.8° subregion (Fig. S2) using these parameter settings.*"

[Figure]

**Fig. R2**. Divisions of the study area. Annual cropland classification is performed within each 0.8°×0.8° subregion. Test regions with a size of 100 km×100 km are used to find the best LandTrendr arguments for each agricultural zone.